# Not All Wrong is Bad: Using Adversarial Examples for Unlearning

**Ali Ebrahimpour-Boroojeny** [1]   **Hari Sundaram** [1]   **Varun Chandrasekaran** [1]

## Abstract

Machine unlearning, where users can request the deletion of a forget dataset, is becoming increasingly important because of numerous privacy regulations. Initial works on "exact" unlearning (e.g., retraining) incur large computational overheads. However, while computationally inexpensive, "approximate" methods have fallen short of reaching the effectiveness of exact unlearning: models produced fail to obtain comparable accuracy and prediction confidence on both the forget and test (i.e., unseen) dataset. Exploiting this observation, we propose a new unlearning method, **A**dversarial **M**achine **UN**learning (AMUN), that outperforms prior state-of-the-art (SOTA) methods for image classification. AMUN lowers the confidence of the model on the forget samples by fine-tuning the model on their corresponding adversarial examples. Adversarial examples naturally belong to the distribution imposed by the model on the input space; fine-tuning the model on the adversarial examples closest to the corresponding forget samples (a) localizes the changes to the decision boundary of the model around each forget sample and (b) avoids drastic changes to the global behavior of the model, thereby preserving the model's accuracy on test samples. Using AMUN for unlearning a random $10\%$ of CIFAR-10 samples, we observe that even SOTA membership inference attacks cannot do better than random guessing.

## 1. Introduction

The goal of *machine unlearning* is to remove the influence of a subset of the training dataset for a model that has been trained on that dataset (Vatter et al., 2023). The necessity for these methods has been determined by privacy regulations such as the European Union's General Data Protection Act

[1]University of Illinois at Urbana-Champaign (UIUC), Illinois, USA. Correspondence to: Ali Ebrahimpour-Boroojeny <ae20@illinois.edu>.

*Proceedings of the $42^{nd}$ International Conference on Machine Learning*, Vancouver, Canada. PMLR 267, 2025. Copyright 2025 by the author(s).

and the California Consumer Privacy Act. Despite early efforts on proposing "exact" solutions to this problem (Cao & Yang, 2015; Bourtoule et al., 2021), the community has favored "approximate" solutions due to their ability to preserve the original model's accuracy while being more computationally efficient (Chen et al., 2023b; Liu et al., 2024; Fan et al., 2024).

Given a training set $\mathcal{D}$ and a subset $\mathcal{D}_F \subset \mathcal{D}$ of the samples that have to be unlearned from a model trained on $\mathcal{D}$, recent works on unlearning have emphasized the use of evaluation metrics that measure the similarity to the behavior of the models that are retrained from scratch on $\mathcal{D} - \mathcal{D}_F$. However, prior unlearning methods do not effectively incorporate this evaluation criterion in the design of their algorithm. In this paper, we first characterize the expected behavior of the retrained-from-scratch models on $\mathcal{D} - \mathcal{D}_F$. Using this characterization, we propose Adversarial Machine UNlearning (AMUN). AMUN is a method that, when applied to the models trained on $\mathcal{D}$, replicates that (desired) behavior after a few iterations. The success of AMUN relies on an intriguing observation: fine-tuning a trained model on the adversarial examples of the training data does not lead to a catastrophic forgetting and instead has limited effect on the deterioration of model's test accuracy.

Upon receiving a request for unlearning a subset $\mathcal{D}_F$ of the training set $\mathcal{D}$, AMUN finds adversarial examples that are *as close as possible* to the samples in $\mathcal{D}_F$. It then utilizes these adversarial examples (with the wrong labels) during fine-tuning of the model for unlearning the samples in $\mathcal{D}_F$. Fine-tuning the model on these adversarial examples, which are naturally mispredicted by the model, decreases the confidence of the predictions on $\mathcal{D}_F$. This decreased confidence of model's predictions on $\mathcal{D}_F$ is similar to what is observed in the models that are retrained on $\mathcal{D} - \mathcal{D}_F$. The distance of these adversarial examples to their corresponding samples in $\mathcal{D}_F$ is much smaller than the distance of $\mathcal{D}_F$ to other samples in $\mathcal{D} - \mathcal{D}_F$; this localizes the effect of fine-tuning to the vicinity of the samples in $\mathcal{D}_F$ and prevents significant changes to the decision boundary of the model and hurting the model's overall accuracy (see § 3.1).

As we will show in § 6, AMUN outperforms prior state-of-the-art (SOTA) unlearning methods (Fan et al., 2024) in unlearning random subsets of the training data from a trained

classification model and closes the gap with the retrained models, even when there is no access to the samples in $\mathcal{D} - \mathcal{D}_\text{F}$ during the unlearning procedure.

To summarize, the main contributions of this work are:

- We observe that neural networks, when fine-tuned on adversarial examples *with their wrong labels*, have limited test accuracy degradation. While prior research in adversarial robustness fine-tune the models on these samples with their labels corrected, we are the first to utilize this form of fine-tuning to get lower prediction confidence scores on the training samples that are present in the proximity of those adversarial examples.

- We introduce a new unlearning method, AMUN, for classification models that outperforms prior methods. It does so by replicating the behavior of the retrained models on the test samples and the forget samples.

- By comparing AMUN to existing unlearning methods using SOTA membership inference attacks (MIAs), we show that it outperforms the other methods in unlearning subsets of training samples of various sizes.

All code can be found in https://github.com/ Ali-E/AMUN

## 1.1. Related Work

Early works in machine unlearning focused on exact solutions (Cao & Yang, 2015; Bourtoule et al., 2021); those ideas were adapted to unlearning in other domains such as graph neural networks (Chen et al., 2022b) and recommendation systems (Chen et al., 2022a). The extensive computational cost and utility loss resulted in the design of approximate methods. An example is the work of Ginart et al. (2019), who provide a definition of unlearning based on differential privacy. Works that followed sought solutions to satisfy those probabilistic guarantees (Ginart et al., 2019; Gupta et al., 2021; Neel et al., 2021; Ullah et al., 2021; Sekhari et al., 2021). However, the methods that satisfy these guarantees were only applied to simple models, such as $k$-means (Ginart et al., 2019) , linear and logistic regression (Guo et al., 2019; Izzo et al., 2021), convex-optimization problems (Neel et al., 2021), or graph neural networks with no non-linearities (Chien et al., 2022). Additional research was carried out to design more scalable approximate methods, those that can be applied to the models that are used in practice, including large neural networks (Golatkar et al., 2020; Warnecke et al., 2021; Izzo et al., 2021; Thudi et al., 2022; Chen et al., 2023b; Liu et al., 2024; Fan et al., 2024). However, these approximate methods do not come with theoretical guarantees; their effectiveness are evaluated using membership inference attacks (MIAs). MIAs aim to determine whether a specific data sample was used in the training set of a trained

model (Shokri et al., 2017; Yeom et al., 2018; Song et al., 2019; Hu et al., 2022; Carlini et al., 2022; Zarifzadeh et al., 2024), and is a common evaluation metric (Liu et al., 2024; Fan et al., 2024). For further discussion on related works, see Appendix B.

## 2. Preliminaries

We begin by introducing the notation we use. We proceed to define various terms in the paper, and conclude by introducing our method.

### 2.1. Notation

Assume a probability distribution $\mathbb{P}_\mathcal{X}$ on the domain of inputs $\mathcal{X}$ and $m$ classes $\mathcal{Y} = \{1, 2, \ldots, m\}$. We consider a multi-class classifier $\mathcal{F} : \mathcal{X} \to \mathcal{Y}$ and its corresponding prediction function $f(x)$ which outputs the probabilities corresponding to each class (e.g., the outputs of the softmax layer in a neural network). The loss function for model $\mathcal{F}$ is denoted $\ell_\mathcal{F} : \mathcal{X} \times \mathcal{Y} \to \mathbb{R}_+$; it uses the predicted scores from $f(x)$ to compute the loss given the true label $y$ (e.g., cross-entropy loss). In the supervised setting we consider here, we are given a dataset $\mathcal{D} = \{(x_i, y_i)\}_{i=\{1, \ldots, N\}}$ that contains labeled samples $x_i \sim \mathbb{P}_\mathcal{X}$ with $y_i \in \mathcal{Y}$. The model $\mathcal{F}$ is trained on $\mathcal{D}$ using the loss $\ell_\mathcal{F}$ to minimize the empirical risk $\mathbb{E}_\mathcal{D}[\ell_\mathcal{F}(x, y)]$ and a set of parameters $\boldsymbol{\theta}_\text{o} \sim \Theta_\mathcal{D}$ is derived for $\mathcal{F}$; $\Theta_\mathcal{D}$ is the distribution over the set of all possible parameters $\Theta$ when the training procedure is performed on $\mathcal{D}$ due to the potential randomness in the training procedure (e.g., initialization and using mini-batch training). We also assume access to a test set $\mathcal{D}_\text{T}$ with samples from the same distribution $\mathbb{P}_\mathcal{X}$. A function $g(x)$ is $L$-Lipschitz if $\|g(x) - g(x')\|_2 \le L\|x - x'\|_2, \forall x, x' \in \mathcal{X}$.

### 2.2. Definitions

**Definition 2.1** (Attack Algorithm). For a given input/output pair $(x, y) \in \mathcal{X} \times \mathcal{Y}$, a model $\mathcal{F}$, and a positive value $\epsilon$, an untargetted attack algorithm $\mathcal{A}_\mathcal{F}(x, \epsilon) = x + \delta_x$ minimizes $\ell_\mathcal{F}(x + \delta_x, y' \ne y)$ such that $\|\delta_x\|_2 \le \epsilon$, where $y' \in \mathcal{Y}$.

**Definition 2.2** (Machine Unlearning). Given the trained model $\mathcal{F}$, and a subset $\mathcal{D}_\text{F} \subset \mathcal{D}$ known as the forget set, the corresponding machine unlearning method is a function $\mathcal{M}_{\mathcal{D}, \mathcal{D}_\text{F}} : \Theta \to \Theta$ that gets $\boldsymbol{\theta}_\text{o} \sim \Theta_\mathcal{D}$ as input and derives a new set of parameters (aka the unlearned model) $\boldsymbol{\theta}_\text{u} \sim \Theta_{\mathcal{D}_\text{F}}$, where $\Theta_{\mathcal{D}_\text{F}}$ is the distribution over the set of parameters when $\mathcal{F}$ is trained on $\mathcal{D} - \mathcal{D}_\text{F}$ rather than $\mathcal{D}$.

### 2.3. Approximate Unlearning

Using Definition 2.2, it is clear that the most straightforward, exact unlearning method would be to retrain model $\mathcal{F}$ from scratch on $\mathcal{D} - \mathcal{D}_\text{F}$; this does not even use $\boldsymbol{\theta}_\text{o}$. However, training deep learning models is very costly, and re-

training the models upon receiving each unlearning request would be impractical. Thus, approximate unlearning methods are designed to overcome these computational requirements by starting from $\theta_\mathrm{o}$ and modifying the parameters to derive $\theta_\mathrm{o}'$ s.t. $\theta_\mathrm{o}' \overset{\mathrm{d}}{=} \theta_\mathrm{u}$ (i.e., from the same distribution).

In the rest of the paper, we refer to $\mathcal{D}_\mathrm{F}$ as the forget or unlearning set interchangeably. Its complement, $\mathcal{D}_\mathrm{R} = \mathcal{D} - \mathcal{D}_\mathrm{F}$ is the remain set. We will use the behavior of the models retrained from scratch on $\mathcal{D}_\mathrm{R}$ as the goal of approximate unlearning methods, and will refer to them as $\mathcal{F}_\mathrm{R}$ for brevity.

### 2.4. Unlearning Settings

Many of the prior methods on approximate unlearning for classification models require access to $\mathcal{D}_\mathrm{R}$. However, in practice, this assumption might be unrealistic. The access to $\mathcal{D}_\mathrm{R}$ might be restricted, or might be against privacy regulations. Prior works do not make a clear distinction based on this requirement when comparing different approximate methods. Therefore, to make a clear and accurate comparison, we perform our experiments (see § 6) in two separate settings: one with access to both $\mathcal{D}_\mathrm{R}$ and $\mathcal{D}_\mathrm{F}$, and the other with access to only $\mathcal{D}_\mathrm{F}$. We report the results for each setting separately. For comparison with prior methods, we adapt them to both settings whenever possible.

## 3. Motivation

We present the intuition for our proposed unlearning method in § 3.1, and in § 3.2 we describe our observation about fine-tuning a model on its adversarial examples.

### 3.1. A Guiding Observation

Before designing a new unlearning method, we would like to first characterize the changes we expect to see after a successful unlearning. Because the retrained models are the gold standard of unlearning methods, we first assess their behavior on $\mathcal{D}_\mathrm{R}$, $\mathcal{D}_\mathrm{F}$, and $\mathcal{D}_\mathrm{T}$. To this end, we evaluate the confidence values of $\mathcal{F}_\mathrm{R}$ when predicting labels of $\mathcal{D}_\mathrm{R}$, $\mathcal{D}_\mathrm{F}$, and $\mathcal{D}_\mathrm{T}$. Since samples in $\mathcal{D}_\mathrm{T}$ are drawn from the same distribution as $\mathcal{D}$, we can conclude that samples in $\mathcal{D}_\mathrm{T}$ and $\mathcal{D}_\mathrm{F}$ are from the same distribution. Therefore, we expect $\mathcal{F}_\mathrm{R}$ to have similar accuracy and prediction confidence scores on $\mathcal{D}_\mathrm{T}$ (test set) and $\mathcal{D}_\mathrm{F}$.

**Results:** Figure 3 in Appendix F.1, shows the confidence scores (see § F.1 for details) for a ResNet-18 (He et al., 2016) model that has been retrained on $\mathcal{D} - \mathcal{D}_\mathrm{F}$, where $\mathcal{D}$ is the training set of CIFAR-10 (Alex, 2009) and the size of $\mathcal{D}_\mathrm{F}$ (randomly chosen from $\mathcal{D}$) is 10% and 50% of the size of $\mathcal{D}$ (the first and second sub-figures, respectively).

> **Key Observation 1:** *The main difference between the predictions on $\mathcal{D}_T$ (unseen samples) and $\mathcal{D}_R$ (observed samples) is that the model's predictions are* much more confident *for the samples that it has observed compared to the unseen samples.*

This basic observation has either been overlooked by the prior research on approximate machine unlearning or has been treated incorrectly. To make the unlearned models more similar to $\mathcal{F}_\mathrm{R}$, prior methods have focused on degrading the model's performance on $\mathcal{D}_\mathrm{F}$ directly by either (a) some variation of fine-tuning on $\mathcal{D}_\mathrm{R}$ (Warnecke et al., 2021; Liu et al., 2024), (b) choosing wrong labels for samples in $\mathcal{D}_\mathrm{F}$ and fine-tuning the model (Golatkar et al., 2020; Chen et al., 2023b; Fan et al., 2024), or (c) directly maximizing the loss with respect to the samples in $\mathcal{D}_\mathrm{F}$ (Thudi et al., 2022). Using the wrong labels for the samples of $\mathcal{D}_\mathrm{F}$ or maximizing the loss on them make these methods very unstable and prone to catastrophic forgetting (Zhang et al., 2024a) because these samples belong to the correct distribution of the data and we cannot force a model to perform wrongly on a portion of the dataset while preserving it's test accuracy. For these methods, it is important to use a small enough learning rate along with early stopping to prevent compromising the model's performance while seeking worse prediction confidence values on the samples in $\mathcal{D}_\mathrm{F}$. Also, most of these methods require access to the set of remaining samples to use it for preventing a total loss of the model's performance (e.g., by continuing to optimize the model on $\mathcal{D}_\mathrm{R}$) (Golatkar et al., 2020; Liu et al., 2024).

### 3.2. Fine-tuning on Adversarial Examples

After training a model $\mathcal{F}$ on $\mathcal{D}$, this model imposes a distribution $f(x)$ (e.g., softmax outputs) for all possible labels $y \in \mathcal{Y}$ given any $x \in \mathcal{X}$. Since the model $\mathcal{F}$ is directly optimized on $\mathcal{D}$, $f(x)$ becomes very skewed toward the correct class for samples in $\mathcal{D}$. For a given sample from $\mathcal{D}$, its adversarial examples (see Definition 2.1) are very close in the input space to the original sample. However, $\mathcal{F}$ makes the wrong prediction on these examples. This wrong prediction is the direct result of the learned parameters $\theta_\mathrm{o}$ for the classification model, and these adversarial examples, although predicted incorrectly, belong to the distribution imposed on $\mathcal{X}$ by these learned parameters (i.e., even though that is not the correct distribution, that is what the model has learned).

Now, what happens if we insert one adversarial example $(x_{adv}, y_{adv})$ that corresponds to the sample $(x, y)$ into $\mathcal{D}$ and make an augmented dataset $\mathcal{D}'$ for fine-tuning? Even before fine-tuning starts, the model makes the correct prediction on that (adversarial) example (by predicting the wrong label $y_{adv}$!), but its confidence might not be as high as the samples in $\mathcal{D}$, on which the model has been trained on. Pro-

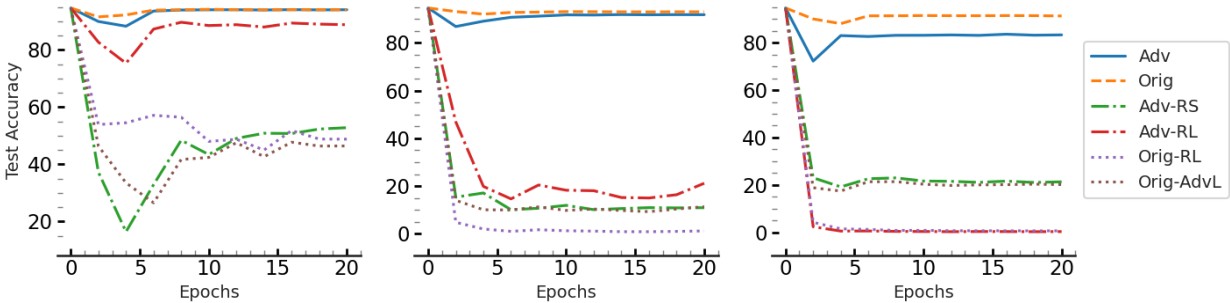

Figure 1: **Effect of fine-tuning on adversarial examples.** This figure shows the effect of fine-tuning on test accuracy of a ResNet-18 model that is trained on CIFAR-10, when the dataset for fine-tuning changes (see § 6.2 for details). Let $\mathcal{D}_F$ contain 10% of the samples in $\mathcal{D}$ and $\mathcal{D}_A$ be the set of adversarial examples constructed using Algorithm 1. Adv, from the left sub-figure to right one, shows the results when $\mathcal{D} \cup \mathcal{D}_A$, $\mathcal{D}_F \cup \mathcal{D}_A$, and $\mathcal{D}_A$ is used for fine-tuning the model, respectively. Orig, Adv-RS, Adv-RL, Orig-RL, and Orig-AdvL shows the results when $\mathcal{D}_A$ for each of these sub-figures is replace by $\mathcal{D}_F$, $\mathcal{D}_{A_{RS}}$, $\mathcal{D}_{A_{RL}}$, $\mathcal{D}_{RL}$, and $\mathcal{D}_{AdvL}$, accordingly. As the figure shows, the specific use of adversarial examples with the mis-predicted labels matters in keeping the model's test accuracy because $\mathcal{D}_A$, in contrast to the other constructed datasets belong to the natural distribution learned by the trained model.

ceeding with fine-tuning of the model on the augmented dataset increases its confidence on $x_{adv}$ while making the same wrong prediction $y_{adv}$. However, this fine-tuning does not change the model's performance because the newly added sample $(x_{adv}, y_{adv})$ does not contradict the distribution learned by the model. Since $(x, y) \in \mathcal{D}'$, and $x$ and $x_{adv}$ are very close to one another (e.g., very similar images) while having different labels, optimizing the model *has to change its decision boundary* in that region of the input space to reach small loss for both of these samples. As a result of this balance, the model tends to decrease its confidence on the original sample compared to the model that was solely trained on $\mathcal{D}$ because there was no opposing components for its optimization on $\mathcal{D}$. Note that $\|x - x_{adv}\| \leq \epsilon$, where $\epsilon$ is often much smaller than the distance of any pairs of samples in $\mathcal{D}$. This helps to localize this change in the decision boundary during fine-tuning, and prevent changes to models' behavior in other regions of the input space (Liang et al., 2023). In the following we elaborate on our empirical observations that verify these changes.

**Setup:** We consider the training set of CIFAR-10 as $\mathcal{D}$ and choose $\mathcal{D}_F$ to be a random subset whose size is 10% of $|\mathcal{D}|$. We also compute a set of adversarial examples (using Algorithm 1) corresponding to $\mathcal{D}_F$, which we call $\mathcal{D}_A$. Fig. 1 shows the fine-tuning of a trained ResNet-18 model for 20 epochs. Similar experiment for VGG19 models trained on the Tiny Imagenet dataset (Le & Yang, 2015) can be found in In the leftmost sub-figure, the curve presented as Orig represents the test accuracy of the model when it is fine-tuned on $\mathcal{D}$. The curve named Adv is fine-tuned on $\mathcal{D} \cup \mathcal{D}_A$, which has a similar test accuracy to Orig.

In the second sub-figure, Orig shows the test accuracy of the model when it is fine-tuned on $\mathcal{D}_F$ (two copies of $\mathcal{D}_F$

to keep the sample count similar), while Adv represents fine-tuning on $\mathcal{D}_F \cup \mathcal{D}_A$. As the figure shows, Adv has a small degradation in test accuracy compared to Orig.

The rightmost sub-figure shows the case where Orig is fine-tuning of the model on $\mathcal{D}_F$, and Adv is fine-tuning on only $\mathcal{D}_A$. Although the degradation in test accuracy increases for this case, surprisingly we see that the model still remains noticeably accurate despite being fine-tuned on a set of samples that are all mislabeled. See § 6.2.1 for more details.

**Results:** As Figure 1 (and Figure 6 and 7 in Appendix F.3) shows, the test accuracy of the model does not deteriorate, even when it is being fine-tuned on only $\mathcal{D}_A$ (the dataset with wrong labels). See § 6.2.1 for further details.

> **Key Observation 2:** *Fine-tuning a model on the adversarial examples does not lead to catastrophic forgetting!*

## 4. Adversarial Machine UNlearning (AMUN)

We utilize our novel observation about the effect of fine-tuning on adversarial examples (see § 3.2) to achieve the intuition we had about the retrained models (see § 3.1). We utilize the existing flaws of the trained model in learning the correct distribution, that appear as adversarial examples in the vicinity of the samples in $\mathcal{D}_F$, to decrease its confidence on those samples while maintaining the performance.

Formally, AMUN uses Algorithm 1 to find an adversarial example for any sample in $(x, y) \in \mathcal{D}_F$. This algorithm uses a given untargeted adversarial algorithm $\mathcal{A}_\mathcal{F}$, that finds the solution to Definition 2.1, for finding an adversarial example

$x_{adv}$. To make sure $\epsilon$ is as small as possible, Algorithm 1 starts with a small $\epsilon$ and runs the attack $\mathcal{A}_{\mathcal{F}}$; if an adversarial algorithm is not found within that radius, it runs $\mathcal{A}_{\mathcal{F}}$ with a larger $\epsilon$. It continues to perform $\mathcal{A}_{\mathcal{F}}$ with incrementally increased $\epsilon$ values until it finds an adversarial example; it then adds it to $\mathcal{D}_A$. The algorithm stops once it finds adversarial examples for all the samples in $\mathcal{D}_{\mathrm{F}}$.

The reason behind minimizing the distance of $\epsilon$ for each sample is to localize the changes to the decision boundary of the model as much as possible; this prevents changing the model's behavior on other parts of the input space. For our experiments, we use PGD-50 (Madry, 2017) with $\ell_2$ norm bound as $\mathcal{A}_{\mathcal{F}}$. We set the step size of the gradient ascent in the attack to $0.1 \times \epsilon$, which changes with the $\epsilon$ value. More details regarding the implementations of AMUN and prior unlearning methods and tuning their hyper-parameters can be found in Appendix C. Also, in Appendix F.5, we will show how using weaker attacks, such as Fast Gradient Sign Method (FGSM) (Goodfellow et al., 2014), might lead to lower performance of AMUN.

---

**Algorithm 1** Build Adversarial Set $(\mathcal{F}, \mathcal{A}, \mathcal{D}_{\mathrm{F}}, \epsilon_{init})$

---

1: **Input:** Model $F$, Attack algorithm $A$, Forget set $\mathcal{D}_{\mathrm{F}}$, and Initial $\epsilon$ for adversarial attack
2: **Output:** $\mathcal{D}_A$: Adversarial set for $\mathcal{D}_{\mathrm{F}}$
3: $\mathcal{D}_A = \{\}$
4: **for** $(x, y)$ **in** $\mathcal{D}_{\mathrm{F}}$ **do**
5:    $\epsilon = \epsilon_{init}$
6:    **while** TRUE **do**
7:       $x_{adv} = \mathcal{A}(x, \epsilon)$
8:       $y_{adv} = \mathcal{F}(x_{adv})$
9:       **if** $y_{adv} ! = y$ **then**
10:          Break
11:       **end if**
12:       $\epsilon = 2\epsilon$
13:    **end while**
14:    Add $(x_{adv}, y_{adv})$ to $\mathcal{D}_A$
15: **end for**
16: **Return** $\mathcal{D}_A$

---

Once Algorithm 1 constructs $\mathcal{D}_A$, AMUN utilizes that to augment the dataset on which it performs the fine-tuning. If $\mathcal{D}_R$ is available, AMUN fine-tunes the model on $\mathcal{D}_R \cup \mathcal{D}_F \cup \mathcal{D}_A$ and when $\mathcal{D}_R$ is not accessible, it performs the fine-tuning on $\mathcal{D}_F \cup \mathcal{D}_A$. Also, in the setting where the size of the $\mathcal{D}_F$ is very large, we noticed some improvement when using only $\mathcal{D}_R \cup \mathcal{D}_A$ and $\mathcal{D}_A$, for those settings, respectively.

### 4.1. Influencing Factors

We also derive an upper-bound on the 2-norm of the difference of the parameters of the unlearned model and the retrained model (which are gold-standard for unlearning)

that illuminated the influencing factors in the effectiveness of AMUN. To prove this theorem, we make assumptions that are common in the certified unlearning literature. The proof is given in Appendix A.

**Theorem 4.1.** *Let $\mathcal{D} = \{(x_i, y_i)\}_{i=\{1,...,N\}}$ be a dataset of $N$ samples and without loss of generality let $(x_n, y_n)$ (henceforth represented as $(x, y)$ for brevity) be the sample that needs to be forgotten and $(x_{adv}, y_{adv})$ be its corresponding adversarial example used by* AMUN *such that $\|x - x_{adv}\|_2 = \delta$. Let $\hat{\mathcal{R}}(w)$ represent the (unnormalized) empirical loss on $\mathcal{D}' = \mathcal{D} \cup \{(x_{adv}, y_{adv})\}$ for a model $f$ that is parameterized with $w$. We assume that $f$ is $L$-Lipschitz with respect to the inputs and $\hat{\mathcal{R}}$ is $\beta$-smooth and convex with respect to the parameters. Let $\theta_o$ represent the parameters corresponding to the model originally trained on $\mathcal{D}$ and $\theta_u$ be the parameters derived when the model is trained on $\mathcal{D} - \{(x, y)\}$. We also assume that both the original and retrained models achieve near-$0$ loss on their training sets. After* AMUN *performs fine-tuning on $\mathcal{D}'$ using one step of gradient descent with a learning rate of $\frac{1}{\beta}$ to derive parameters $\theta'$, we get the following upper-bound for the distance of the unlearned model and the model retrained on $\mathcal{D} - \{(x, y)\}$ (gold standard of unlearning):*

$$\|\theta' - \theta_u\|_2^2 \leq \|\theta_o - \theta_u\|_2^2 + \frac{2}{\beta}(L\delta - C),$$

*where $C = \ell(f_{\theta_o}(x_{adv}), y) + \ell(f_{\theta'}(x_{adv}), y_{adv}) - \ell(f_{\theta_u}(x), y) - \ell(f_{\theta_u}(x_{adv}), y_{adv})$.*

According to the bound in Theorem 4.1, a lower Lipschitz constant of the model ($L$) and adversarial examples that are closer to the original samples (lower value for $\delta$) lead to a smaller upper bound. A larger value of $C$ also leads to a improved upper-bound. In the following we investigate the factors that lead to a larger value for $C$, which further clarifies some of influencing factors in the effectiveness of AMUN:

- Higher quality of adversarial example in increasing the loss for the correct label on the original model, which leads to larger value for $\ell(f_{\theta_o}(x_{adv}), y)$.

- Transferability of the adversarial example generated on the original model to the retrained model to decrease its loss for the wrong label, which leads to a lower value for $\ell(f_{\theta_u}(x_{adv}), y_{adv})$. This also aligns with lower Lipschitz constant of the model, as shown by prior work (Ebrahimpour-Boroojeny et al.).

- Early stopping and using appropriate learning rate during fine-tuning phase of unlearning to avoid overfitting to the adversarial example, which does not allow low values for $\ell(f_{\theta'}(x_{adv}), y_{adv})$.

- The generalization of the retrained model to the unseen samples, which leads to a lower value for $\ell(f_{\theta_u}(x), y)$.

Note that the first two implications rely on the strength of the adversarial example in addition to being close to the original sample. The second bullet, which relies on the transferability of adversarial examples, has been shown to improve as the Lipschitz constant decreases (Ebrahimpour-Boroojeny et al., 2024). The third bullet point is a natural implication which also holds for other unlearning methods that rely on the fine-tuning of the model. The fourth bullet point is not relevant to the unlearning method and instead relies on the fact that the retrained model should have good generalizability to unseen samples; it implies that as the size of $\mathcal{D}_F$ increases (i.e., $|\mathcal{D}_R|$ decreases) and the performance of the retrained model decreases, the effectiveness of the unlearning model also decreases. This is also intuitively expected in the unlearning process. Hence, the proved theorem also justifies our earlier intuitions about the need for good quality adversarial examples that are as close as possible to the original samples (which is the goal of Algorithm 1).

## 5. Evaluation Setup

In this section we elaborate on the details of evaluating different unlearning methods. More details (e.g., choosing the hyper-parameters) can be found in Appendix C.

### 5.1. Baseline Methods

We compare AMUN with `FT` (Warnecke et al., 2021), `RL` (Golatkar et al., 2020), `GA` (Thudi et al., 2022), `BS` (Chen et al., 2023b), $l_1$-Sparse (Liu et al., 2024), and `SalUn` (Fan et al., 2024). We also combine the weight saliency idea for masking the model parameters to limit the changes to the parameters during fine-tuning with AMUN and present its results as $\text{AMUN}_{+SalUn}$ (see Appendix D for more details). We use the same hyper-parameter tuning reported by prior works. For further details, see Appendix C.

### 5.2. Evaluation Metrics

The metic used by recent works in unlearning to evaluate the unlearning methods (Liu et al., 2024; Fan et al., 2024) considers the models retrained on $\mathcal{D}_R$ as the goal standard for comparison. They compute the following four values for both the retrained models and the models unlearned using approximate methods:

- *Unlearn Accuracy:* Their accuracy on $\mathcal{D}_F$.

- *Retain Accuracy:* Their accuracy on $\mathcal{D}_R$.

- *Test Accuracy:* Their accuracy on $\mathcal{D}_T$.

- *MIA score*: Scores returned by membership inference attacks on $\mathcal{D}_F$

Once these four values are computed, the absolute value of the difference of each of them with the corresponding value for $\mathcal{F}_R$ (the retrained models) is computed. Finally, the average of the four differences (called the *Average Gap*) is used as the metric to compare the unlearning methods.

The MIAs used in the recent unlearning methods by Liu et al. (2024); Fan et al. (2024) are based on the methods introduced by Yeom et al. (2018); Song et al. (2019). Although these MIAs have been useful for basic comparisons, recent SOTA MIAs significantly outperform their earlier counterparts, albeit with an increase in complexity and computation cost. To perform a comprehensive comparison of the effectiveness of the unlearning methods, we utilized a SOTA MIA called RMIA (Zarifzadeh et al., 2024), in addition to using the MIAs from prior works. In RMIA, the area under the ROC curve (AUC) of the MIA scores for predicting the training samples from the unseen samples is reported. Recall that in machine unlearning, the samples are split to three sets: $\mathcal{D}_R$, $\mathcal{D}_F$, and $\mathcal{D}_T$. For an unlearning method to be effective, as discussed in § 3.1, we expect the AUC of RMIA for distinguishing the samples in $\mathcal{D}_F$ from the ones in $\mathcal{D}_T$ to be the same as random guessing (50% assuming balanced data). As shown in Table 1, this expectation holds for the models retrained on $\mathcal{D}_R$.

We report the results of our comparisons for both the MIAs from prior unlearning literature and the new SOTA MIA. We will present the former one as `MIS`, and the latter one as `FT AUC` (the AUC of predicting $\mathcal{D}_F$ from $\mathcal{D}_T$).

### 5.3. Unlearning Settings

Another important factor missing in the comparisons of the unlearning methods in prior works is the possibility of access to $\mathcal{D}_R$. So, for our experiments we consider two settings, one with access to $\mathcal{D}_R$ and one with access to only $\mathcal{D}_F$. We adapt each of the unlearning methods to both of these settings, and perform the comparisons in each of these settings separately. The prior unlearning methods that do not adapt to the setting where there is no access to $\mathcal{D}_R$ (Warnecke et al., 2021; Liu et al., 2024) are excluded for the presented results in that setting.

Therefore, we perform different sets of experiments to evaluate the unlearning methods in both settings, and hope this becomes the norm in future works in machine unlearning. In each of these two settings, we evaluate unlearning of 10% or 50% of the samples randomly chosen from $\mathcal{D}$. For all the experiments we train three models on $\mathcal{D}$. For each size of $\mathcal{D}$, we use three random subsets and for each subset, we try three different runs of the unlearning methods. This leads to a total of 27 runs of each unlearning method using different initial models and subsets of $\mathcal{D}$ to unlearn.

# 6. Experiments

We wish to answer the following questions:

1. Does AMUN lead to effective unlearning of any random subset of the samples when evaluated by a SOTA MIA?

2. Does the choice of $\mathcal{D}_A$ matter in AMUN, or can it be replaced with a dataset that contains different labels or different samples that are within the same distance to the corresponding samples in $\mathcal{D}_F$?

3. Is AMUN effective on adversarially robust models?

4. Does the choice of attack method matter in Algorithm 1 used by AMUN and does transferred attack work as well?

5. How does AMUN compare to other unlearning methods when used for performing multiple unlearning requests on the same model?

As a quick summary, our results show that: (1) AMUN effectively leads to unlearning the samples in $\mathcal{D}_F$: after unlearning $10\%$ of the samples of CIFAR-10 from a trained ResNet-18, RMIA cannot do better than random guessing (§ 6.1); (2) If we replace $\mathcal{D}_A$ with any of the aforementioned substitutes, the model's accuracy significantly deteriorates, especially when there is no access to $\mathcal{D}_R$ (§ 6.2.1); (3) AMUN is as effective for unlearning on models that are adversarially robust (§ 6.2.2); (4) using weaker attack methods, such as FGSM, in AMUN hurts the effectiveness by not finding the adversarial examples that are very close to the samples in $\mathcal{D}_F$. However, they still outperform prior methods (Appendix F.5). The transferred adversarial examples are effective as well (Appendix F.4); and (5) AMUN outperforms other unlearning methods when handling multiple unlearning requests (§ 6.3).

## 6.1. Effectiveness of AMUN

In this subsection we report the results on the comparisons of AMUN to other unlearning methods (see § 5.1). We consider the unlearning settings discussed in § 5.3, and the evaluation metrics discussed in § 5.2. We use ResNet-18 models trained on CIFAR-10 and VGG19 models trained on Tiny Imagenet for this experiment. We also perform an analysis on the computation costs of AMUN (see section E.1 for the details).

**Results:** Table 1 shows the results of evaluation using RMIA when the unlearning methods *have access to* $\mathcal{D}_R$. Table 2 shows these results when there is *no access to* $\mathcal{D}_R$. As the results show, AMUN *clearly outperforms prior unlearning methods in all settings*. This becomes even more clear when there is no access to $\mathcal{D}_R$. Note that, for the

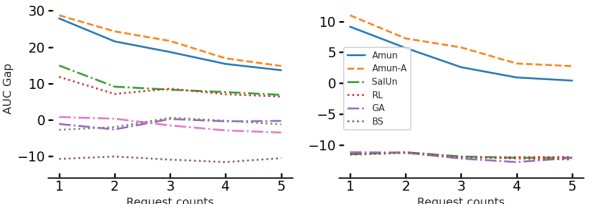

Figure 2: **Multiple unlearning requests.** This figure shows the evaluation of unlearning methods when they are used for unlearning for five times and each time on $2\%$ of the training data. We train a ResNet-18 model on CIFAR-10 when $\mathcal{D}_R$ is available (left) and when it is not (right). After each step of the unlearning, we use the MIA scores generated by RMIA to derive the area under the ROC curve (AUC) for $\mathcal{D}_R$ vs. $\mathcal{D}_F$ and $\mathcal{D}_F$ vs. $\mathcal{D}_T$. The values on the y-axis shows the difference of these two AUC scores. A high value for this gap means the samples in $\mathcal{D}_F$ are far more similar to $\mathcal{D}_T$ rather than $\mathcal{D}_R$ and shows a more effective unlearning.

models retrained on $\mathcal{D}_R$, the AUC score of RMIA for predicting $\mathcal{D}_R$ from $\mathcal{D}_T$ (which can be considered as the worst case for `FT AUC` score) are 64.17 and 69.05 for unlearning $10\%$ and $50\%$ accordingly. Similar results for unlearning in VGG19 models trained on Tiny Imagenet when unlearning $10\%$ of $\mathcal{D}$ can be found in section E.

We also present the results when MIS is used as the evaluation metric in Tables 10 and 11 in Appendix G, which similarly shows AMUN's dominance in different unlearning settings. Moreover, we evaluate the combination of AMUN and `SalUn` (see Appendix D for details) and present its results as AMUN$_{SalUn}$ in these tables. AMUN$_{SalUn}$ slightly improves the results of AMUN in the setting where there is no access to $\mathcal{D}_R$, by filtering the parameters that are more relevant to $\mathcal{D}_F$ during fine-tuning.

## 6.2. Ablation Studies

In this subsection, we first elaborate on the effect of fine-tuning a model on its adversarial examples and compare it to the cases where either the samples or labels of this dataset change (§ 6.2.1). We then discuss AMUN's efficacy on models that are already robust to adversarial examples (§ 6.2.2). We present other ablation studies on using weaker, but faster, adversarial attacks in Algorithm 1 (Appendix F.5). In Appendix F.4, we utilize transferred adversarial examples for unlearning, as this can expedite handling the unlearning from a newly trained model for which adversarial examples on similar architectures are available.

### 6.2.1. FINE-TUNING ON ADVERSARIAL EXAMPLES

We want to verify the importance of $\mathcal{D}_A$ (created by Algorithm 1) in preserving the model's test accuracy. To this end, we build multiple other sets to be used instead of $\mathcal{D}_A$ when

| | RANDOM FORGET (10%) | | | | | RANDOM FORGET (50%) | | | | |
|---|---|---|---|---|---|---|---|---|---|---|
| | UNLEARN ACC | RETAIN ACC | TEST ACC | FT AUC | AVG. GAP | UNLEARN ACC | RETAIN ACC | TEST ACC | FT AUC | AVG. GAP |
| RETRAIN | 94.49 ±0.20 | 100.0 ±0.00 | 94.33 ±0.18 | 50.00 ±0.42 | 0.00 | 92.09 ±0.37 | 100.0 ±0.00 | 91.85 ±0.33 | 50.01 ±0.12 | 0.00 |
| FT | 95.16 ±0.29 | 96.64 ±0.25 | 92.21 ±0.27 | 52.08 ±0.34 | 2.06 ±0.10 | 94.24 ±0.30 | 95.82 ±0.31 | 91.21 ±0.33 | 51.74 ±0.36 | 2.17 ±0.13 |
| RL | 95.54 ±0.14 | 97.47 ±0.08 | 92.17 ±0.10 | 51.33 ±0.63 | 1.74 ±0.18 | 94.83 ±0.44 | 99.79 ±0.04 | 90.08 ±0.16 | 50.78 ±0.14 | 1.38 ±0.09 |
| GA | 98.94 ±1.39 | 99.22 ±1.31 | 93.39 ±1.18 | 60.96 ±2.93 | 4.28 ±0.47 | 100.00 ±0.00 | 100.00 ±0.00 | 94.65 ±0.07 | 63.39 ±0.26 | 4.62 ±0.05 |
| BS | 99.14 ±0.31 | 99.89 ±0.06 | 93.04 ±0.14 | 57.85 ±1.12 | 3.48 ±0.32 | 55.24 ±5.11 | 55.67 ±4.90 | 50.16 ±5.28 | 55.19 ±0.42 | 32.01 ±3.86 |
| $l_1$-SPARSE | 94.29 ±0.34 | 95.63 ±0.16 | 91.55 ±0.17 | 51.21 ±0.32 | 2.16 ±0.06 | 98.00 ±0.17 | 98.71 ±0.13 | 92.79 ±0.10 | 54.44 ±0.47 | 2.67 ±0.11 |
| SALUN | 96.25 ±0.21 | 98.14 ±0.16 | 93.06 ±0.18 | 50.88 ±0.54 | 1.44 ±0.12 | 96.68 ±0.35 | 99.89 ±0.01 | 91.97 ±0.18 | 50.86 ±0.18 | 1.36 ±0.04 |
| AMUN | 95.45 ±0.19 | 99.57 ±0.00 | 93.45 ±0.22 | 50.18 ±0.36 | **0.62** ±0.05 | 93.50 ±0.09 | 99.71 ±0.01 | 92.39 ±0.04 | 49.99 ±0.18 | **0.33** ±0.03 |
| AMUN$_{+SalUn}$ | 95.02 ±0.18 | 99.58 ±0.04 | 93.29 ±0.04 | 50.72 ±0.79 | 0.68 ±0.18 | 93.56 ±0.07 | 99.72 ±0.02 | 92.52 ±0.20 | 49.81 ±0.40 | 0.36 ±0.07 |

Table 1: **Unlearning with access to $\mathcal{D}_\mathbf{R}$.** Comparing different unlearning methods in unlearning 10% and 50% of $\mathcal{D}$. Avg. Gap (see § 5.2) is used for evaluation (lower is better). The lowest value is shown in bold while the second best is specified with underscore. As the results show, AMUN outperforms all other methods by achieving lowest Avg. Gap and AMUN$_{SalUn}$ achieves comparable results.

| | RANDOM FORGET (10%) | | | | | RANDOM FORGET (50%) | | | | |
|---|---|---|---|---|---|---|---|---|---|---|
| | UNLEARN ACC | RETAIN ACC | TEST ACC | FT AUC | AVG. GAP | UNLEARN ACC | RETAIN ACC | TEST ACC | FT AUC | AVG. GAP |
| RETRAIN | 94.49 ±0.20 | 100.0 ±0.00 | 94.33 ±0.18 | 50.00 ±0.42 | 0.00 | 92.09 ±0.37 | 100.0 ±0.00 | 91.85 ±0.33 | 50.01 ±0.12 | 0.00 |
| RL | 100.00 ±0.00 | 100.00 ±0.00 | 94.45 ±0.09 | 61.85 ±0.25 | 4.31 ±0.06 | 100.00 ±0.00 | 100.00 ±0.00 | 94.57 ±0.14 | 61.99 ±0.10 | 4.29 ±0.03 |
| GA | 4.77 ±3.20 | 5.07 ±3.54 | 5.09 ±3.38 | 49.78 ±0.34 | 68.53 ±2.45 | 100.00 ±0.00 | 100.00 ±0.00 | 92.65 ±0.09 | 63.41 ±0.24 | 5.13 ±0.04 |
| BS | 100.00 ±0.00 | 100.00 ±0.00 | 94.48 ±0.04 | 61.41 ±0.29 | 4.20 ±0.07 | 100.00 ±0.00 | 100.00 ±0.00 | 94.58 ±0.08 | 62.43 ±0.14 | 4.40 ±0.05 |
| SALUN | 100.00 ±0.00 | 100.00 ±0.00 | 94.47 ±0.10 | 61.09 ±0.40 | 4.11 ±0.09 | 100.00 ±0.00 | 100.00 ±0.00 | 94.59 ±0.12 | 62.45 ±0.37 | 4.40 ±0.07 |
| AMUN | 94.28 ±0.37 | 97.47 ±0.10 | 91.67 ±0.04 | 52.24 ±0.23 | 1.94 ±0.13 | 92.77 ±0.52 | 95.66 ±0.25 | 89.43 ±0.19 | 52.60 ±0.22 | 2.51 ±0.09 |
| AMUN$_{+SalUn}$ | 94.19 ±0.38 | 97.71 ±0.06 | 91.79 ±0.12 | 51.93 ±0.12 | **1.77** ±0.06 | 91.90 ±0.63 | 96.59 ±0.31 | 89.98 ±0.44 | 52.32 ±0.56 | **2.00** ±0.17 |

Table 2: **Unlearning with access to only $\mathcal{D}_\mathbf{F}$.** Comparing different unlearning methods in unlearning 10% and 50% of $\mathcal{D}$. Avg. Gap (see § 5.2) is used for evaluation (lower is better) when only $\mathcal{D}_\mathrm{F}$ is available during unlearning. As the results show, AMUN$_{SalUn}$ significantly outperforms all other methods, and AMUN achieves comparable results.

fine-tuning. Let us assume that $\mathcal{A}_\mathcal{F}(x,y) = (x_{adv}, y_{adv})$. Then, these other sets are:

- $\mathcal{D}_{AdvL}$: $\{(x, y_{adv})\}_{\forall(x,y)\in\mathcal{D}_\mathrm{F}}$
- $\mathcal{D}_{RL}$ : $\{(x, y'), \text{s.t. } y' \neq y, y_{adv}\}_{\forall(x,y)\in\mathcal{D}_\mathrm{F}}$
- $\mathcal{D}_{ARL}$: $\{(x_{adv}, y'), \text{s.t.} y' \neq y, y_{adv}\}_{\forall(x,y)\in\mathcal{D}_\mathrm{F}}$
- $\mathcal{D}_{ARS}$: $\{(x', y_{adv}), \text{s.t. } x' \sim \text{Uniform}(X_\delta), \text{where } X_\delta = \{\forall \hat{x} : \|x_\delta - x\|_2 = \delta\}\}_{\forall(x,y)\in\mathcal{D}_\mathrm{F}}$

In this experiment, we evaluate the effect of fine-tuning on test accuracy of a ResNet-18 model that is trained on CIFAR-10, when $\mathcal{D}_\mathrm{A}$ is substituted with other datasets that vary in the choice of samples or their labels. We assume that $\mathcal{D}_\mathrm{F}$ contains 10% of the samples in $\mathcal{D}$ and $\mathcal{D}_\mathrm{A}$ is the set of corresponding adversarial examples constructed using Algorithm 1.

**Results:** In Fig. 1, Adv, from the left sub-figure to the right sub-figure, shows the results when $\mathcal{D} \cup \mathcal{D}_\mathrm{A}$, $\mathcal{D}_\mathrm{F} \cup \mathcal{D}_\mathrm{A}$, and $\mathcal{D}_\mathrm{A}$ is used for fine-tuning the model, respectively. Orig, Adv-RS, Adv-RL, Orig-RL, and Orig-AdvL show the results when $\mathcal{D}_\mathrm{A}$ for each of these sub-figures is replaced by $\mathcal{D}_\mathrm{F}$, $\mathcal{D}_{ARS}$, $\mathcal{D}_{ARL}$, $\mathcal{D}_{RL}$, and $\mathcal{D}_{AdvL}$, respectively. As the figure shows, the specific use of adversarial examples with the mispredicted labels matters in keeping the model's

test accuracy, especially as we move from the leftmost sub-figure (having access to $\mathcal{D}_\mathrm{R}$) to the right one (only using $\mathcal{D}_\mathrm{A}$ or its substitutes). This is due to the fact that the samples in $\mathcal{D}_\mathrm{A}$, in contrast to the other constructed datasets, belong to the natural distribution learned by the trained model. Therefore, even if we only fine-tune the ResNet-18 model on $\mathcal{D}_\mathrm{A}$, we still do not lose much in terms of model's accuracy on $\mathcal{D}_\mathrm{T}$. This is a surprising observation, as $\mathcal{D}_\mathrm{A}$ contains a set of samples with wrong predictions! Fig. 6 in Appendix F.3 shows similar results when size of $\mathcal{D}_\mathrm{F}$ is 50% of $|\mathcal{D}|$.

### 6.2.2. ADVERSARIALLY ROBUST MODELS

We evaluate the effectiveness of AMUN when the trained model is adversarially robust. One of the most effective methods in designing robust models is adversarial training which targets smoothing the model's prediction function around the training samples (Salman et al., 2019). This has been shown to provably enhance the adversarial robustness of the model (Cohen et al., 2019). One of the effective adversarial training methods is by using TRADES loss introduced by (Zhang et al., 2019). We will use adversarially trained ResNet-18 models for unlearning 10% of the samples in CIFAR-10. In addition, we will use another defense mechanism that is less costly and more practical for larger models.

There is a separate line of work that try to achieve the same smoothness in model's prediction boundary by controlling the Lipschitz constant of the models (Szegedy, 2013). The method proposed by Boroojeny et al. (2024) is much faster than adversarial training and their results show a significant improvement in the robust accuracy. We use their clipping method to evaluate the effectiveness of AMUN for unlearning 10% and 50% of the samples from robust ResNet-18 models trained on CIFAR-10.

**Results:** Table 7 in Appendix F.2 shows the results for the adversarially trained models. For the models with controlled Lipschitz continuity, the results are shows in Table 3 (no access to $\mathcal{D}_\text{R}$) and Table 6 in Appendix F.2 (with access to $\mathcal{D}_\text{R}$). As the results show, even when there is no access to $\mathcal{D}_\text{R}$, AMUN still results in effective unlearning for adversarially robust models; RMIA does not do better than random guessing for predicting $\mathcal{D}_\text{F}$ from $\mathcal{D}_\text{T}$. As Fig. 3 (right) shows, in the robust models, more than 97% of the adversarial examples are further away from their corresponding training samples, compared to this distance for the original models. However, this does not interfere with the performance of AMUN because these robust models are smoother and tend to be more regularized. This regularization, which prevents them from overfitting to the training samples is in fact a contributing factor to the improved generalization bounds for these models (Bartlett et al., 2017). This in itself contributes to enhanced resilience against MIAs. As seen in Tables 3 and 6, even for the clipped models retrained on $\mathcal{D}_\text{R}$, the AUC score of RMIA for predicting $\mathcal{D}_\text{R}$ from $\mathcal{D}_\text{F}$ (FR AUC) is very low, which shows that these smoother models .

| | **Random Forget** (10%) | | | **Random Forget** (50%) | | |
|---|---|---|---|---|---|---|
| | FT AUC | FR AUC | Test Acc | FT AUC | FR AUC | Test Acc |
| Retrain | 49.95 ±0.24 | 54.08 ±0.16 | 89.01 ±0.21 | 50.19 ±0.15 | 55.61 ±0.05 | 85.76 ±0.41 |
| AMUN | 49.55 ±0.13 | 54.01 ±0.23 | 87.55 ±0.44 | 49.64 ±0.31 | 53.23 ±0.21 | 87.39 ±0.61 |

Table 3: **Unlearning on adversarially robust models.** Evaluating the effectiveness of AMUN in unlearning 10% and 50% of the training samples when the models are adversarially robust and there is no access to $\mathcal{D}_\text{R}$. For this experiment we use models with controlled Lipschitz constant which makes them provably and empirically more robust to adversarial examples.

### 6.3. Continuous Unlearning

We evaluate the performance of the unlearning methods when they are used to perform multiple consecutive unlearning from a trained model. This is a desirable capability for unlearning methods because in real-world applications there might be multiple unlearning requests and it is preferred to minimize the number of times that a model needs to be retrained from scratch. The setting we envision is as follows: models are updated at each request for unlearning. For AMUN, this means that $\mathcal{D}_\text{A}$ is computed on an updated model after each set of unlearning requests (shown as AMUN-A). In addition to comparing AMUN-A to the other unlearning methods, we also compare it to a version (shown as AMUN) that computes all the adversarial examples on the original model so it can handle the unlearning requests faster upon receiving them i.e., $\mathcal{D}_\text{A}$ is not computed on an updated model after each request; the set of requests are batched and $\mathcal{D}_\text{A}$ is computed on the entire batch. For this experiment, we use a ResNet-18 model trained on training set of CIFAR-10 (50K samples). Our goal is to unlearn 10% of the training samples (5K), but this time in 5 consecutive sets of size 2% (1K) each. We then evaluate the effectiveness of unlearning at each step using RMIA.

**Results:** Fig. 2 shows an overview of the results for both settings of unlearning (with or without access to $\mathcal{D}_\text{R}$). This figure shows the effectiveness of unlearning by depicting how the samples in $\mathcal{D}_\text{F}$ are more similar to the test samples ($\mathcal{D}_\text{T}$) rather than the remaining samples ($\mathcal{D}_\text{R}$). The value on the y-axis shows the difference of the area under the ROC curve (AUC) for predicting $\mathcal{D}_\text{R}$ from $\mathcal{D}_\text{F}$ and $\mathcal{D}_\text{F}$ from $\mathcal{D}_\text{T}$. For the plots of each of these values separately, see Appendix H. AMUN-A performs better than all the other unlearning methods for all the steps of unlearning. Although AMUN also outperforms all the prior unlearning methods, it slightly under-performs compared to AMUN-A. This is expected, as the model's decision boundary slightly changes after each unlearning request and the adversarial examples generated for the original model might not be as effective as those ones generated for the new model. Note that for this experiment, we did not perform hyper-parameter tuning for any of the unlearning methods, and used the same ones derived for unlearning 10% of the dataset presented in § 6.1. For further discussion of the results see Appendix H.

## 7. Conclusions

AMUN utilizes our new observation on how fine-tuning the trained models on adversarial examples that correspond to a subset of the training data does not lead to significant deterioration of model's accuracy. Instead, it decreases the prediction confidence values on the the corresponding training samples. By evaluating AMUN using SOTA MIAs, we show that it outperforms other existing method, especially when unlearning methods do not have access to the remaining samples. It also performs well for handling multiple unlearning requests. This work also raises some questions for future work: (1) Since SOTA MIA methods fail to detect the unlearned samples, can this method be used to provide privacy guarantees for all the training samples?; (2) Can the same ideas be extended to generative models or Large Language Models?; (3) Can we derive theoretical bounds on the utility loss due to fine-tuning on adversarial examples?

# Acknowledgments

This work used Delta computing resources at National Center for Supercomputing Applications through allocation CIS240316 from the Advanced Cyberinfrastructure Coordination Ecosystem: Services & Support (ACCESS) program (Boerner et al., 2023), which is supported by U.S. National Science Foundation grants #2138259, #2138286, #2138307, #2137603, and #2138296. This paper was generously supported by NSF award IIS-2312561.

# Impact Statement

This research advances machine unlearning, a critical capability for privacy compliance in AI systems. Traditional exact unlearning methods, such as retraining, are computationally expensive, while approximate methods struggle to maintain model accuracy and confidence. AMUN introduces a novel approach that efficiently lowers model confidence on forget samples by leveraging adversarial examples, ensuring targeted changes to the decision boundary without significantly altering overall model behavior. This breakthrough improves privacy protection by making membership inference attacks ineffective while maintaining test accuracy, setting a new standard for efficient, privacy-preserving unlearning in deep learning. The work paves the way for scalable and effective unlearning solutions, addressing a fundamental challenge in AI regulation and ethical machine learning.

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

# Appendix

## A. Proofs

Here we provide the proof of Theorem 4.1:

*Proof.* As we perform the unlearning by fine-tuning and performing a gradient descent update to $\theta_o$, we have: $\theta' = \theta_o - \frac{1}{\beta}\nabla\hat{\mathcal{R}}(\theta_o)$. Therefore, we can write:

$$
\begin{aligned}
\|\theta' - \theta_u\|_2^2 &= \|\theta_o - \frac{1}{\beta}\nabla\hat{\mathcal{R}}(\theta_o) - \theta_u\|_2^2 \\
&= \|\theta_o - \theta_u\|_2^2 - \frac{2}{\beta}\langle\nabla\hat{\mathcal{R}}(\theta_o), \theta_o - \theta_u\rangle + \frac{1}{\beta^2}\|\nabla\hat{\mathcal{R}}(\theta_o)\|_2^2 \\
&\leq \|\theta_o - \theta_u\|_2^2 + \frac{2}{\beta}(\hat{\mathcal{R}}(\theta_u) - \hat{\mathcal{R}}(\theta_o)) + \frac{2}{\beta}(\hat{\mathcal{R}}(\theta_o) - \hat{\mathcal{R}}(\theta')) \\
&= \|\theta_o - \theta_u\|_2^2 + \frac{2}{\beta}(\hat{\mathcal{R}}(\theta_u) - \hat{\mathcal{R}}(\theta')),
\end{aligned}
$$

where the inequality is derived by using the smoothness property ($\|\nabla\hat{\mathcal{R}}(\theta_o)\|_2^2 \leq 2\beta(\hat{\mathcal{R}}(\theta_o) - \hat{\mathcal{R}}(\theta'))$) and the convexity assumption which leads to the inequality: $\hat{\mathcal{R}}(\theta_o)) \geq \hat{\mathcal{R}}(\theta_u) + \langle\nabla\hat{\mathcal{R}}(\theta_o), \theta_o - \theta_u\rangle$.

Next, we derive an upper-bound for $\hat{\mathcal{R}}(\theta_u) - \hat{\mathcal{R}}(\theta')$ to replace in the above inequality. By the definition of unnormalized empirical loss on $\mathcal{D}'$:

$$
\begin{aligned}
&\hat{\mathcal{R}}(\theta_u) - \hat{\mathcal{R}}(\theta') \\
&= \sum_{i=1}^{n-1}\ell(f_{\theta_u}(x_i), y_i) + \ell(f_{\theta_u}(x), y) + \ell(f_{\theta_u}(x'), y') - \sum_{i=1}^{n-1}\ell(f_{\theta'}(x_i), y_i) - \ell(f_{\theta'}(x), y) - \ell(f_{\theta'}(x'), y') \\
&= \ell(f_{\theta_u}(x), y) + \ell(f_{\theta_u}(x'), y') - \ell(f_{\theta'}(x), y) - \ell(f_{\theta'}(x'), y'),
\end{aligned}
$$

where the last equality was derived by the assumption that models are trained until they achieve near-0 loss on their corresponding dataset. Therefore, $\sum_{i=1}^{n-1}\ell(f_{\theta_u}(x_i), y_i) = \sum_{i=1}^{n-1}\ell(f_{\theta'}(x_i), y_i) = 0$ since the retrained model has been trained on the remaining samples and the unlearned model has been derived by a single step of gradient descent on the original model, that had been trained on $\mathcal{D}$.

To further simplify the derived terms above and reaching at our desired inequality, we focus on the term $-\ell(f_{\theta'}(x), y)$. By adding and decreasing the term $\ell(f_{\theta_o}(x'), y)$ we get:

$$
\begin{aligned}
-\ell(f_{\theta'}(x), y) &= -\ell(f_{\theta_o}(x'), y) + \ell(f_{\theta_o}(x'), y) - \ell(f_{\theta'}(x), y) \\
&\leq -\ell(f_{\theta_o}(x'), y) + \ell(f_{\theta_o}(x'), y) - \ell(f_{\theta_o}(x), y) - \langle\nabla\ell(f_{\theta_o}(x), y), \theta' - \theta_o\rangle \\
&= -\ell(f_{\theta_o}(x'), y) + \ell(f_{\theta_o}(x'), y) - \ell(f_{\theta_o}(x), y) \\
&\leq -\ell(f_{\theta_o}(x'), y) + L\delta,
\end{aligned}
$$

where the first inequality uses the convexity of the the loss function with respect to the parameters and the third derivations is due to the assumption that the original model achieves a zero loss on its training samples, including $(x, y)$ (hence, $\nabla\ell(f_{\theta_o}(x), y) = 0$). The final inequality is due to the Lipschitzness assumption of model $f$ with respect to the inputs.

$\square$

## B. Related Works (cont.)

To the best of our knowledge AMUN is the first work that considers fine-tuning of a model on the adversarial examples with their wrong labels as a method for unlearning a subset of the samples. However, upon reviewing the prior works in unlearning literature, there are several works that their titles might suggest otherwise. Therefore, here we mention a few of these methods and how they differ from our work.

To improve upon fine-tuning on samples in $\mathcal{D}_{\text{F}}$ with randomly chosen wrong labels, Chen et al. (2023b) use the labels derived from one step of the FGSM attack to choose the new labels for the samples in $\mathcal{D}_{\text{F}}$. This method which was presented as `BS` in our experiments (§ 6.1), does not use the adversarial examples and only uses their labels as the new labels for samples of $\mathcal{D}_{\text{F}}$. This corresponds to the dataset $\mathcal{D}_{AdvL}$ in § 6.2.1. As our results in Figures 1 and 6 show, fine-tuning the trained model on this dataset leads to catastrophic forgetting even when $\mathcal{D}_{\text{R}}$ is available. This is simply due to the fact that the samples in $\mathcal{D}_{AdvL}$ contradict the distribution that the trained models have already learned.

The work by Setlur et al. (2022) is not an unlearning method, despite what the name suggest. They propose a regularization method that tries to maximize the loss on the adversarial examples of the training samples that are relatively at a higher distance to lower the confidence of the model on those examples. The work by Zhang et al. (2024b) proposed a defense method similar to adversarial training for making to unlearned LLMs more robust to jailbreak attacks on the topics that they have unlearned. Łucki et al. (2024) also study the careful application of jailbreak attacks against unlearned models. The work by Jung et al. (2024) investigate computing adversarial noise to mask the model parameters. Many of the works with similar titles, use "adversarial" to refer to minimax optimization (Zeng et al., 2021) or considering a Stackelberg game setting between the source model and the adversary that is trying to extract information (Di et al., 2024).

## C. Implementation Details

For all the experiments we train three models on $\mathcal{D}$. For each size of $\mathcal{D}$ ($10\%$ or $50\%$), we use three random subsets and for each subset, we try three different runs of each of the unlearning methods. This leads to a total of 27 runs of each unlearning method using different initial models and subsets of $\mathcal{D}$ to unlearn. Hyper-parameter tuning of each of the methods is done on a separate random subset of the same size from $\mathcal{D}$, and then the average performance is computed for the other random subsets used as $\mathcal{D}_{\text{F}}$. For tuning the hyper-parameters of the models, we followed the same range suggested by their authors and what has been used in the prior works for comparisons. Similar to prior works (Liu et al., 2024; Fan et al., 2024), we performed 10 epochs for each of the unlearning methods, and searched for best learning rate and number of steps for a learning rate scheduler. More specifically, for each unlearning method, we performed a grid search on learning rates within the range of $[10^{-6}, 10^{-1}]$ with an optional scheduler that scales the learning rate by $0.1$ for every 1 or 5 steps. For `SalUn`, whether it is used on its own or in combination with AMUN, we searched for the masking ratios in the range $[0.1, 0.9]$.

The original models are ResNet-18 models trained for 200 epochs with a learning rate initialized at 0.1 and using a scheduler that scales the learning rate by 0.1 every 40 epochs. The retrained models are trained using the same hyper-parameters as the original models. For evaluation using RMIA, we trained 128 separate models such that each sample is included in half of these models. As suggested by the authors, we used Soft-Margin Taylor expansion of Softmax (SM-Taylor-Softmax) with a temperature of 2 for deriving the confidence values in attacks of RMIA. We used the suggested threshold of 2 for comparing the ratios in computing the final scores ($\gamma$ value). For controlling the Lipschitz constant of the ResNet-18 models in § 6.2.2, we used the default setting provided by the authors for clipping the spectral norm of all the convolutional and fully-connected layers of the model to 1. For RMIA evaluations, we trained 128 of these models separately such that each sample appears in exactly half of these models.

## D. AMUN + SalUn

The main idea behind SalUn is to limit the fine-tuning of the model, during unlearning, to only a subset of the parameters of the model, while keeping the rest of them fixed. Fan et al. (2024) show that this technique helps to preserve the accuracy of the model when fine-tuning the model on $\mathcal{D}_{\text{F}}$ with randomly-chosen wrong labels. More specifically, they compute a mask using the following equation:

$$\mathbf{m}_{\text{S}} = \mathbf{1}\left(|\nabla_{\boldsymbol{\theta}_{\text{o}}} \ell(\boldsymbol{\theta}_{\text{o}}; \mathcal{D}_{\text{F}})\,|| \geq \gamma\right),$$

which, basically, computes the gradient of the loss function for the current parameters with respect to $\mathcal{D}_{\text{F}}$, and uses threshold $\gamma$ to filter the ones that matter more to the samples in $\mathcal{D}_{\text{F}}$. Note that, $\mathbf{1}$ is an element-wise indicator function. Then, during

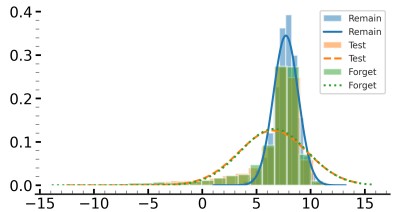 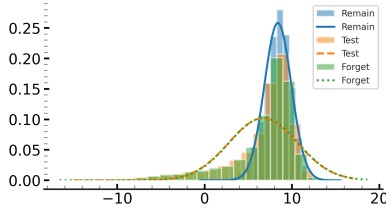 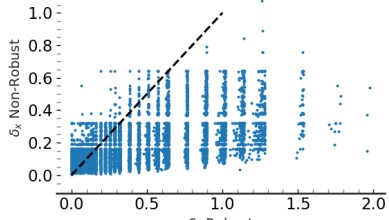

Figure 3: (left) These two plots show the histogram of confidence values of the retrained model on its predictions for the remaining set (Remain), test set (Test), and forget set (Forget) during the training, when the size of the forget set is $\%10$ (1st plot) and $\%50$ (2nd plot) of the training set. It also shows the Gaussian distributions fitted to each histogram. As the plots show the models perform similarly on the forget set and test set because to the retrained model they are unseen data from the same distribution. (right) This plot compares the $\delta_x$ value in definition 2.1 for adversarial examples generated on the original ResNet-18 models (x-axis) and clipped ResNet-18 models (y-axis). The dashed line shows $x = y$ line and more than $97\%$ of the values fall bellow this line.

fine-tuning of the model on $\mathcal{D}_F$ with random labels they use $\mathbf{m}_S$ to detect the parameters of $\boldsymbol{\theta}_o$ that get updated.

In our experiments, we try combining this idea with AMUN for updating a subset of the parameters that might be more relevant to the samples in $\mathcal{D}_F$. We refer to this combination as AMUN$_{SalUn}$ in Tables 1 and 2 in § 6.1 and Tables 10 and 11 in Appendix G. As the results show, AMUN$_{SalUn}$ constantly outperforms SalUn and for the cases that $\mathcal{D}_R$ is not available it also outperforms AMUN. In the setting where $\mathcal{D}_R$ is accessible, it performs comparable to AMUN. This is probably due to the fact that when $\mathcal{D}_R$ is not available and AMUN has access to only the samples $\mathcal{D}_F \cup \mathcal{D}_A$, SalUn acts as a regularization for not allowing all the parameters of the model that might not be relevant to $\mathcal{D}_F$ be updated. In the setting where $\mathcal{D}_R$ is available, involving it in fine-tuning will be a sufficient regularization that preserves models' utility while unlearning $\mathcal{D}_F$.

## E. Effectiveness of AMUN (cont.)

In this subsection we report the results on the comparisons of AMUN to other unlearning methods (see § 5.1) for unlearning $10\%$ of the training samples for VGG19 models that are trained on Tiny Imagenet dataset. We consider the unlearning settings discussed in § 5.3, and the evaluation metrics discussed in § 5.2. Table 4 shows the results of evaluation using RMIA when the unlearning methods *have access to* $\mathcal{D}_R$. Table 5 shows these results when there is *no access to* $\mathcal{D}_R$. As the results show, AMUN *clearly outperforms prior unlearning methods in all settings*. This becomes even more clear when there is no access to $\mathcal{D}_R$.

| | UNLEARN ACC | RETAIN ACC | TEST ACC | FT AUC | AVG. GAP |
|---|---|---|---|---|---|
| RETRAIN | $55.93_{\pm 0.21}$ | $99.98_{\pm 0.00}$ | $55.96_{\pm 0.74}$ | $50.06_{\pm 0.36}$ | $0.00$ |
| FT | $74.99_{\pm 1.13}$ | $96.88_{\pm 0.48}$ | $56.29_{\pm 0.47}$ | $61.66_{\pm 0.36}$ | $8.36_{\pm 0.37}$ |
| RL | $59.66_{\pm 1.77}$ | $68.19_{\pm 1.55}$ | $50.78_{\pm 0.95}$ | $51.38_{\pm 0.34}$ | $10.50_{\pm 0.07}$ |
| GA | $0.46_{\pm 0.01}$ | $0.49_{\pm 0.02}$ | $0.50_{\pm 0.02}$ | $49.80_{\pm 0.08}$ | $52.67_{\pm 0.03}$ |
| BS | $0.50_{\pm 0.11}$ | $0.51_{\pm 0.01}$ | $0.51_{\pm 0.02}$ | $49.80_{\pm 0.02}$ | $52.65_{\pm 0.04}$ |
| $l_1$-SPARSE | $55.27_{\pm 1.41}$ | $60.64_{\pm 1.24}$ | $49.82_{\pm 0.72}$ | $54.49_{\pm 0.20}$ | $12.80_{\pm 0.52}$ |
| SALUN | $66.54_{\pm 4.61}$ | $76.63_{\pm 3.83}$ | $49.56_{\pm 1.19}$ | $54.64_{\pm 0.52}$ | $11.24_{\pm 0.14}$ |
| **AMUN** | $62.57_{\pm 0.62}$ | $93.66_{\pm 0.66}$ | $55.52_{\pm 0.67}$ | $57.33_{\pm 0.14}$ | $\underline{5.17}_{\pm 0.13}$ |
| **AMUN**$_{+SalUn}$ | $62.96_{\pm 0.92}$ | $94.42_{\pm 0.49}$ | $55.80_{\pm 0.55}$ | $57.65_{\pm 0.46}$ | $\mathbf{5.09}_{\pm 0.29}$ |

Table 4: **Unlearning with access to $\mathcal{D}_R$.** Comparing different unlearning methods in unlearning $10\%$ of Tiny Imagenet Dataset ($\mathcal{D}$) from VGG19 models. Avg. Gap is used for evaluation (lower is better). The lowest value is shown in bold while the second best is specified with underscore. As the results show, AMUN outperforms all other methods by achieving lowest Avg. Gap and AMUN$_{SalUn}$ achieves comparable results.

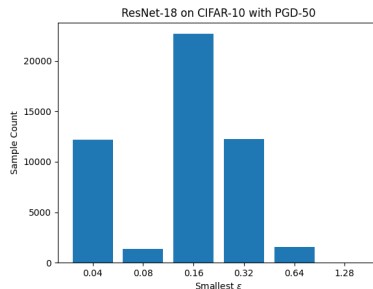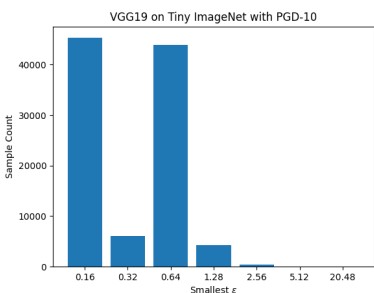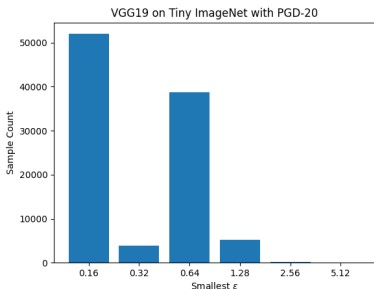

Figure 4: Each plot shows the portion of samples for final values of $\epsilon$ in Algorithm 1. The smallest value on the $x$-axis in each plot shows the initial $\epsilon$ value chosen for Algorithm 1 based on running this algorithm on a small subset of the dataset. The left-most plot shows this distribution for CIFAR-10 when the chosen attack algorithm is PGD-50. The two right-most plots show the distributions for the Tiny ImageNet dataset when PGD-10 and PGD-20 are used, respectively. Once the initial value is set, a run of PGD attack is performed on all the samples. For the ones that adversarial example is not found within that radius, we perform other runs of PGD attack until adversarial examples are found for all the samples. As the histograms show, the total work is equivalent to less than 3 runs of PGD attack on the whole dataset, which is not computationally expensive.

### E.1. Computational Cost

One option to fast processing of the unlearning request is to run Algorithm 1 on all the training samples after the model is trained; this allows access to $\mathcal{D}_\text{A}$ for any arbitrary $\mathcal{D}_\text{F}$ provided. The alternative, is to run Algorithm 1 on $\mathcal{D}_\text{F}$ once the unlearning request is received. In either of these two cases, note that Algorithm 1 can be run in parallel on all the samples starting from the initial value for $\epsilon$. Then for only the samples that the adversarial example is not found, we perform another iteration of attack with the updated value for $\epsilon$. We continue this procedure until all the samples of interest have a corresponding adversarial example in $\mathcal{D}_\text{A}$. Choosing a reasonable initial value for $\epsilon$ can save the computation time, by avoiding initial iterations on almost all the samples without any outcome. To choose the initial value of $\epsilon$ we run Algorithm 1 with a very small $\epsilon$ on a small subset of $\mathcal{D}_\text{F}$ (e.g., 100 samples). Then we choose $\epsilon_{init}$ such that at least 95% of the samples find their adversarial examples within that distance. Using this strategy, we find that even when running Algorithm 1 on all $\mathcal{D}$, the computation time will be equivalent to running the underlying adversarial attack (e.g., PGD-50) less than 3 times on all the samples in $\mathcal{D}$. Figure 4 shows the histogram for the number of samples in $\mathcal{D}$ for each $\epsilon$ value for different models and datasets. The smallest value of $\epsilon$ in these plots is the $\epsilon_{init}$ chosen by the sampling procedure mentioned earlier. Note that a large portion of the samples find their corresponding adversarial examples within the first few iterations.

|  | Unlearn Acc | Retain Acc | Test Acc | FT AUC | Avg. Gap |
|---|---|---|---|---|---|
| Retrain | $55.93$ $_{\pm 0.21}$ | $99.98$ $_{\pm 0.00}$ | $55.96$ $_{\pm 0.74}$ | $50.06$ $_{\pm 0.36}$ | $0.00$ |
| RL | $1.36$ $_{\pm 0.21}$ | $1.74$ $_{\pm 0.34}$ | $1.27$ $_{\pm 0.18}$ | $51.17$ $_{\pm 0.42}$ | $52.15$ $_{\pm 0.07}$ |
| GA | $0.44$ $_{\pm 0.05}$ | $0.51$ $_{\pm 0.01}$ | $0.50$ $_{\pm 0.00}$ | $50.55$ $_{\pm 0.97}$ | $52.81$ $_{\pm 0.14}$ |
| BS | $0.93$ $_{\pm 0.24}$ | $0.97$ $_{\pm 0.22}$ | $1.02$ $_{\pm 0.27}$ | $49.96$ $_{\pm 0.20}$ | $52.28$ $_{\pm 0.19}$ |
| SalUn | $2.46$ $_{\pm 1.51}$ | $2.89$ $_{\pm 1.68}$ | $2.13$ $_{\pm 0.98}$ | $52.20$ $_{\pm 0.94}$ | $51.63$ $_{\pm 0.82}$ |
| **AMUN** | $58.54$ $_{\pm 1.56}$ | $62.34$ $_{\pm 1.40}$ | $43.22$ $_{\pm 0.68}$ | $63.05$ $_{\pm 0.89}$ | $\underline{16.50}$ $_{\pm 0.13}$ |
| **AMUN**$_{+SalUn}$ | $62.37$ $_{\pm 0.61}$ | $68.47$ $_{\pm 0.70}$ | $45.44$ $_{\pm 0.72}$ | $61.24$ $_{\pm 0.63}$ | $\mathbf{14.91}$ $_{\pm 0.37}$ |

Table 5: **Unlearning without access to $\mathcal{D}_\mathbf{R}$.** Comparing different unlearning methods in unlearning 10% of Tiny Imagenet Dataset ($\mathcal{D}$) from VGG19 models. Avg. Gap is used for evaluation (lower is better). The lowest value is shown in bold while the second best is specified with underscore. As the results show, AMUN outperforms all other methods by achieving lowest Avg. Gap and AMUN$_{SalUn}$ achieves comparable results.

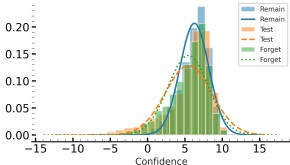 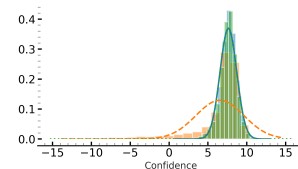 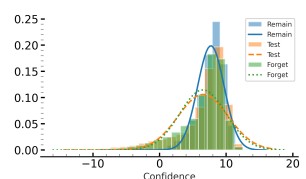

Figure 5: The two left-most subplots show the confidence values before and after unlearning (using AMUN) of $10\%$ of the training samples. The two right-most subplots show these confidence values for unlearning $50\%$ of the training samples. In both cases, the confidence values of samples in $\mathcal{D}_F$ are similar to those of $\mathcal{D}_R$ and their fitted Gaussian distribution matches as expected. After using AMUN for unlearning the samples in $\mathcal{D}_F$, the confidence values on this set gets more similar to the test (unseen) samples.

## F. Ablation Study (cont.)

In this section, we further discuss the ablation studies that were mentioned in § 6.2. We also present other ablation studies on using transferred adversarial examples (Appendix F.4) and weaker adversarial attacks (Appendix F.5) in AMUN.

### F.1. Empirical Behavior of Retrained Models

As discussed in § 3.1, assuming the $\mathcal{D}_T$ and $\mathcal{D}$ come from the same distributions, we expect the prediction confidences of the models retrained on $\mathcal{D}_R$ to be similar on $\mathcal{D}_F$ and $\mathcal{D}_T$, because both of these sets are considered unseen samples that belong the the same data distribution. Figure 3 (left) shows the confidence scores for a ResNet-18 model that has been retrained on $\mathcal{D} - \mathcal{D}_F$, where $\mathcal{D}$ is the training set of CIFAR-10 and the size of $\mathcal{D}_F$ (randomly chosen from $\mathcal{D}$) is $10\%$ and $50\%$ of the size of $\mathcal{D}$ for the left and right sub-figures, respectively. To derive the confidence values, we use the following scaling on the logit values:

$$\phi(f(x)_y) = \log\left(\frac{f(x)_y}{1 - f(x)_y}\right),$$

where $f(x)_y$ is the predicted probability for the correct class. This scaling has been used by Carlini et al. (2022) to transform the the prediction probabilities such that they can be better approximated with a normal distribution, which are indeed used by some of the SOTA MIA methods for predicting training samples from the test samples (Carlini et al., 2022). Figure 3 (left) shows these fitted normal distribution as well, which perfectly match for $\mathcal{D}_F$ and $\mathcal{D}_T$.

#### F.1.1. CONFIDENCE VALUES IN UNLEARNED MODELS

In this section, we investigate the confidence values of the model, before and after using AMUN for unlearning a subset of $10\%$ or $25\%$ of the training samples. For the original model (before unlearning), we expect the distribution of confidence values of samples in $\mathcal{D}_F$ to be similar to those of the samples in $\mathcal{D}_R$ because they were both used as the training data and the model has used them similarly during training. However, this distribution is different for the test samples ($\mathcal{D}_T$), as the model has not seen them during the training phase. After unlearning, as discussed in section 3.1, we expect the distribution of confidence values for $\mathcal{D}_F$ and $\mathcal{D}_T$ to become more similar so that MIAs cannot distinguish them from each other. As Figure 5 shows, for both unlearning $10\%$ (two leftmost subplots) and $50\%$ (two rightmost subplots), we observe the same behavior. Fur the original models (1st and 3rd subplot), the distribution for $\mathcal{D}_F$ and $\mathcal{D}_R$ mathces exactly, but after using AMUN (2nd and 4th subplot) the distribution for $\mathcal{D}_F$ shifts toward that of $\mathcal{D}_T$.

### F.2. Adversarially Robust Models (cont.)

As discussed in § 6.2.2, we also evaluatee the effectiveness of AMUN when the trained model is adversarially robust. For this experiment, we used the ResNet-18 models with 1-Lipschitz convolutional and fully-connected layers, which are shown to be significantly more robust than the original ResNet-18 models. In Table 3, we showed the results for unlearning $10\%$ and $50\%$ of the samples from the robust ResNet-18 models trained on CIFAR-10, in the case where $\mathcal{D}_R$ is not accessible. In Table 6, we showed the corresponding results when the unlearning methods have access to $\mathcal{D}_R$. As the results show, similar to the results discussed in § 6.2.2, AMUN effectively unlearns $\mathcal{D}_F$ for either of the sizes of the this set.

|  | RANDOM FORGET (10%) | | | RANDOM FORGET (50%) | | |
|---|---|---|---|---|---|---|
|  | FT AUC | FR AUC | TEST ACC | FT AUC | FR AUC | TEST ACC |
| RETRAIN | $49.95_{\pm 0.24}$ | $54.08_{\pm 0.16}$ | $89.01_{\pm 0.21}$ | $50.19_{\pm 0.15}$ | $55.61_{\pm 0.05}$ | $85.76_{\pm 0.41}$ |
| AMUN | $49.12_{\pm 0.19}$ | $53.60_{\pm 0.31}$ | $86.94_{\pm 0.56}$ | $49.41_{\pm 0.25}$ | $54.22_{\pm 0.16}$ | $87.38_{\pm 0.39}$ |

Table 6: **Unlearning on adversarially robust models.** Evaluating the effectiveness of AMUN in unlearning $10\%$ and $50\%$ of the training samples when the models are adversarially robust and we have access to $\mathcal{D}_\text{R}$. For this experiment we use models with controlled Lipschitz constant which makes them provably and empirically more robust to adversarial examples.

We also evaluated AMUN for unlearning in models that are adversarially trained. we performed our analysis on ResNet-18 models trained using TRADES loss (Zhang et al., 2019) on CIFAR-10. We performed the experiments for unlearning $10\%$ of the dataset in both cases where $\mathcal{D}_\text{R}$ is accessible and not. As the results in Table 7 show, in both settings AMUN is effective in unlearning the forget samples and achieving a low gap with the retrained models. This gap is obviously smaller when there is access to $\mathcal{D}_\text{R}$.

|  | UNLEARN ACC | RETAIN ACC | TEST ACC | FT AUC | AVG. GAP |
|---|---|---|---|---|---|
| RETRAIN | $82.33_{\pm 0.39}$ | $94.22_{\pm 0.21}$ | $81.72_{\pm 0.36}$ | $50.04_{\pm 0.34}$ | $0.00$ |
| AMUN $_{\text{With }\mathcal{D}_\text{R}}$ | $82.65_{\pm 0.62}$ | $94.33_{\pm 0.84}$ | $84.99_{\pm 0.91}$ | $47.18_{\pm 0.50}$ | $1.02_{\pm 0.18}$ |
| AMUN $_{\text{No }\mathcal{D}_\text{R}}$ | $81.38_{\pm 0.10}$ | $87.45_{\pm 0.54}$ | $79.74_{\pm 0.31}$ | $54.61_{\pm 0.23}$ | $3.57_{\pm 0.24}$ |

Table 7: **Unlearning with access to $\mathcal{D}_\textbf{R}$.** Evaluating AMUN when applied to ResNet-18 models trained using adversarial training. TRADES loss is used to train the models, and the unlearning is done on $10\%$ of CIFAR-10 Dataset ($\mathcal{D}$). Avg. Gap is used for evaluation (lower is better). The result has been reported in two cases: with and without access to $\mathcal{D}_\text{R}$. As the results show, AMUN is effective in both cases, with slight degradation in the more difficult setting of no access to $\mathcal{D}_\text{R}$.

### F.3. Fine-tuning on Adversarial Examples (cont.)

As explained in § 6.2.1, we evaluate the effect of fine-tuning on test accuracy of a ResNet-18 model that is trained on CIFAR-10, when $\mathcal{D}_\text{A}$ is substituted with other datasets that vary in the choice of samples or their labels (see § 6.2.1 for details). In Figure 1 we presented the results when $\mathcal{D}_\text{F}$ contains $10\%$ of the samples in $\mathcal{D}$. We also present the results for the case where $\mathcal{D}_\text{F}$ contains $50\%$ of the samples in $\mathcal{D}$ in Figure 6. As the figure shows, even for the case where we fine-tune the trained models on only $\mathcal{D}_\text{A}$ which contains the adversarial examples corresponding to $50\%$ of the samples in $\mathcal{D}$ (right-most sub-figure), there is no significant loss in models' accuracy. This is due to the fact that the samples in $\mathcal{D}_\text{A}$, in contrast to the other constructed datasets, belong to the natural distribution learned by the trained model. To generate the results in both Figures 1 and 6, we fine-tuned the trained ResNet-18 models on the all the datasets (see § 6.2.1 for details) for 20 epochs. We used a learning rate of $0.01$ with a scheduler that scales the learning rate by $0.1$ every 5 epochs.

We also perform the same experiment for VGG19 (Simonyan & Zisserman, 2014) models trained on Tiny Imagenet dataset (Le & Yang, 2015). We evaluate the effect of fine-tuning on test accuracy of these model that is, when $\mathcal{D}_\text{F}$ contains $10\%$ of the samples in $\mathcal{D}$ and $\mathcal{D}_\text{A}$ is substituted with other datasets that vary in the choice of samples or their labels (see § 6.2.1 for details). In Figure 7 we presented the results, which similarly show that the specific use of adversarial examples with the mis-predicted labels matters in keeping the model's test accuracy.

### F.4. Transferred Adversarial Examples

One of the intriguing properties of adversarial attacks is their transferability to other models (Papernot et al., 2016; Liu et al., 2016); Adversarial examples generated on a trained model (source model) mostly transfer successfully to other models (target models). This success rate of the transferred adversarial examples increases if the source model and target model have the same architecture (Papernot et al., 2016). There are other studies that can be used to increase the success rate of this type of attack (Zhao et al., 2021; Zhang et al., 2022; Chen et al., 2023a; Ebrahimpour-Boroojeny et al., 2024). In this section, we are interested to see if using the the adversarial examples generated using Algorithm 1 for a given model

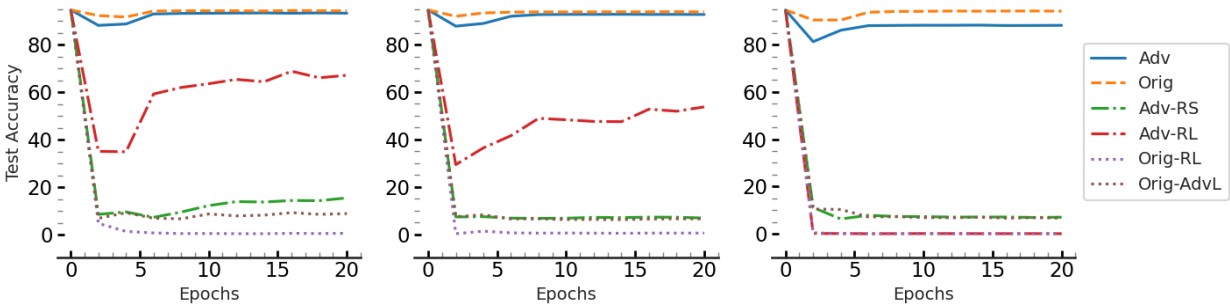

Figure 6: This figure shows the effect of fine-tuning on test accuracy of a ResNet-18 model that is trained on CIFAR-10, when the dataset for fine-tuning changes (see § 6.2 for details). Let $\mathcal{D}_F$ contain 50% of the samples in $\mathcal{D}$ and $\mathcal{D}_A$ be the set of adversarial examples constructed using Algorithm 1. Adv, from the left sub-figure to right one, shows the results when $\mathcal{D} \cup \mathcal{D}_A$, $\mathcal{D}_F \cup \mathcal{D}_A$, and $\mathcal{D}_A$ is used for fine-tuning the model, respectively. Orig, Adv-RS, Adv-RL, Orig-RL, and Orig-AdvL shows the results when $\mathcal{D}_A$ for each of these sub-figures is replace by $\mathcal{D}_F$, $\mathcal{D}_{ARS}$, $\mathcal{D}_{ARL}$, $\mathcal{D}_{RL}$, and $\mathcal{D}_{AdvL}$, accordingly. As the figure shows, the specific use of adversarial examples with the mis-predicted labels matters in keeping the model's test accuracy because $\mathcal{D}_A$, in contrast to the other constructed datasets belong to the natural distribution learned by the trained model.

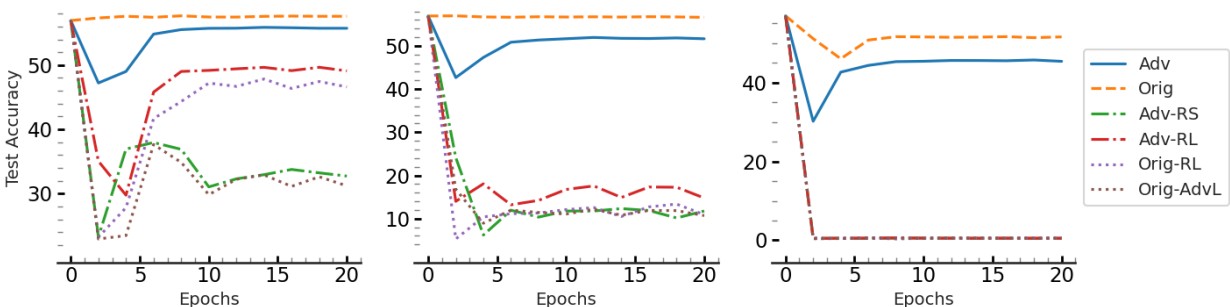

Figure 7: **Effect of fine-tuning on adversarial examples.** This figure shows the effect of fine-tuning on test accuracy of a VGG19 model that is trained on the Tiny ImagenNet dataset, when the dataset for fine-tuning changes for details). Let $\mathcal{D}_F$ contain 10% of the samples in $\mathcal{D}$ and $\mathcal{D}_A$ be the set of adversarial examples constructed using Algorithm 1. Adv, from the left sub-figure to right one, shows the results when $\mathcal{D} \cup \mathcal{D}_A$, $\mathcal{D}_F \cup \mathcal{D}_A$, and $\mathcal{D}_A$ is used for fine-tuning the model, respectively. Orig, Adv-RS, Adv-RL, Orig-RL, and Orig-AdvL shows the results when $\mathcal{D}_A$ for each of these sub-figures is replace by $\mathcal{D}_F$, $\mathcal{D}_{ARS}$, $\mathcal{D}_{ARL}$, $\mathcal{D}_{RL}$, and $\mathcal{D}_{AdvL}$, accordingly. As the figure shows, the specific use of adversarial examples with the mis-predicted labels matters in keeping the model's test accuracy because $\mathcal{D}_A$, in contrast to the other constructed datasets belong to the natural distribution learned by the trained model.

trained on some dataset $\mathcal{D}$ can be used as the $\mathcal{D}_A$ dataset for unlearning a portion of $\mathcal{D}$ from a separately trained model. The advantage of using adversarial examples generated for another model is saving the computation cost for other trained models. For this purpose, we train three ResNet-18 models separately on CIFAR-10, we generate the adversarial examples for each of these models using Algorithm 1. We use AMUN for unlearning 10% and 50% of CIFAR-10 from either of these models, but instead of their adversarial samples, we use the ones derived from the other models. The results in Table 8 shows that using transferred adversarial examples leads to lower performance, specially for the case where there is no access to $\mathcal{D}_R$. All the values for test accuracy are also lower compared to using adversarial examples from the model itself because these adversarial examples from the other models do not all belong to the natural distribution of the model and they do not even always transfer to the other models. Still the results are comparable to the prior SOTA methods in unlearning, and even in the case of no access to $\mathcal{D}_R$ outperforms all prior methods.

| | WITH ACCESS TO $\mathcal{D}_R$ | | | | | | NO ACCESS TO $\mathcal{D}_R$ | | | | | |
| | RANDOM FORGET (10%) | | | RANDOM FORGET (50%) | | | RANDOM FORGET (10%) | | | RANDOM FORGET (50%) | | |
| | TEST ACC | FT AUC | AVG. GAP | TEST AUC | FT AUC | AVG. GAP | TEST ACC | FT AUC | AVG. GAP | TEST AUC | FT AUC | AVG. GAP |
|---|---|---|---|---|---|---|---|---|---|---|---|---|
| **SELF** | 93.45 $\pm$0.22 | 50.18 $\pm$0.36 | 0.62 $\pm$0.05 | 92.39 $\pm$0.04 | 49.99 $\pm$0.18 | 0.33 $\pm$0.03 | 91.67 $\pm$0.04 | 52.24 $\pm$0.23 | 1.94 $\pm$0.13 | 89.43 $\pm$0.19 | 52.60 $\pm$0.22 | 2.51 $\pm$0.09 |
| **OTHERS** | 92.64 $\pm$0.09 | 48.70 $\pm$0.59 | 1.57 $\pm$0.12 | 91.49 $\pm$0.03 | 47.36 $\pm$0.63 | 1.15 $\pm$0.23 | 90.56 $\pm$0.28 | 48.29 $\pm$0.22 | 3.07 $\pm$0.15 | 83.61 $\pm$0.45 | 51.11 $\pm$0.04 | 6.70 $\pm$0.33 |

Table 8: **Transferred adversarial examples.** Comparing the effectiveness of unlearning when instead of using adversarial examples of the model, we use adversarial examples generated using Algorithm 1 on separately trained models with the same architecture. As the results show, relying on transferred adversarial examples in AMUN leads to worse results, specially for test accuracy because the adversarial examples do not necessary belong to the natural distribution learned by the model. However, even by using these transferred adversarial examples AMUN outperforms prior SOTA unlearning methods, specially when there is no access to $\mathcal{D}_R$.

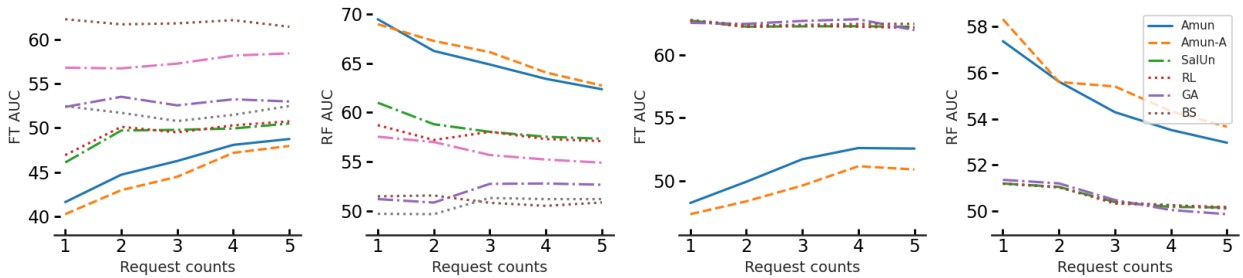

Figure 8: This figure shows both `FT AUC` and `RF AUC` components of the plots presented in Figure 2. The two left-most sub-figures show these values along the number of unlearning requests for the case where there is access to $\mathcal{D}_R$ and the two right-most ones show these values when there is no access to $\mathcal{D}_R$.

## F.5. Weak Attacks

In this section we evaluate the effectiveness of using weaker attacks in Algorithm 1. For this purpose, we perform the unlearning on a ResNet-18 model trained on CIFAR-10 in all unlearning settings mentioned in § 5, and compare the results with the default choice of PGD-50 in AMUN. The weaker attack that we use is a variation of FFGSM (Wong et al., 2020), which itself is a variant of FGSM (Goodfellow et al., 2014). FGSM takes steps toward the gradient sign at a given sample to find adversarial samples. FFGSM takes a small step toward a random direction first, and then proceeds with FGSM. To adapt these method to the format of Algorithm 1 we start with FGSM attack; we find the gradient sign and start to move toward that direction in steps of size $\epsilon$ until we find an adversarial example. If the adversarial example is not found after a few iteration of the `While` loop, we restart the value of $\epsilon$ and add a small random perturbation before the next round of FGSM attack and the `While` loop. We continue this procedure to find an adversarial sample. After deriving a new set of adversarial examples using this methods, we performed a separate round hyper-parameter tuning for unlearning with the new attack to have a fair comparison. It is notable to mention that this leads to a much faster attack because we only compute the gradient once for each round round of FGSM (at the beginning or after each addition of random perturbation and restarting FGSM). Table 9 shows the comparison of the results with the original version of AMUN that uses PGD-50. As the results show, using this weaker attack leads to worse results; however, they still outperform prior SOTA methods in unlearning, specially in the setting where there is no access to $\mathcal{D}_R$ and the size of $\mathcal{D}_F$ is 50% of $\mathcal{D}$.

For each image in CIFAR-10, Figure 9 shows $\delta_x$ (see Definition 2.1) for the adversarial examples that Algorithm 1 finds using PGD-50 ($x$-axis) and FFGSM ($y$-axis). The dashed line shows the $x = y$ line for the reference. As the figure shows $\delta_x$ is much smaller for PGD-50. This value is smaller for FFGSM for less than 4% of the images, but still even for those images, the value of $\delta_x$ for PGD-50 is very small, compared to the range of values that are required for FFGSM in many cases. This, we believe, is the main reason behind worse performance when using FFGSM. However, still note that the adversarial examples that are found using FFGSM belong to the natural distribution of the trained model and therefore fine-tuning the model on these samples does not lead to noticable deterioration of the test accuracy, while achieving reasonable `FT AUC` score. Indeed this larger distance of the adversarial examples with the original samples in $\mathcal{D}_F$, leads to better performance of AMUN when it does not include $\mathcal{D}_F$ when fine-tuning the model, because the difference in the predicted logits compared to the $\delta_x$ leads to under-estimation of the local Lipschitz constant and therefore, the model is able to fit perfectly to both the original samples and its corresponding adversarial sample without changing much. This consequently leads to a larger value

of `FT AUC` score.

| | WITH ACCESS TO $\mathcal{D}_R$ | | | | | | NO ACCESS TO $\mathcal{D}_R$ | | | | | |
| | RANDOM FORGET (10%) | | | RANDOM FORGET (50%) | | | RANDOM FORGET (10%) | | | RANDOM FORGET (50%) | | |
| | TEST ACC | FT AUC | AVG. GAP | TEST AUC | FT AUC | AVG. GAP | TEST ACC | FT AUC | AVG. GAP | TEST AUC | FT AUC | AVG. GAP |
|---|---|---|---|---|---|---|---|---|---|---|---|---|
| **PGD-50** | 93.45 ±0.22 | 50.18 ±0.36 | 0.62 ±0.05 | 92.39 ±0.04 | 49.99 ±0.18 | 0.33 ±0.03 | 91.67 ±0.04 | 52.24 ±0.23 | 1.94 ±0.13 | 89.43 ±0.19 | 52.60 ±0.22 | 2.51 ±0.09 |
| **FGSM** | 93.87 ±0.16 | 50.64 ±0.51 | 0.92 ±0.25 | 89.41 ±1.01 | 50.93 ±0.46 | 1.81 ±0.77 | 92.14 ±0.28 | 56.58 ±1.05 | 3.46 ±0.36 | 90.12 ±0.28 | 54.54 ±0.47 | 3.29 ±0.10 |

Table 9: **Using weaker attacks.** Comparing the effectiveness of unlearning when PGD-10 in Algorithm 1 is replaced with a variant of FGSM attack, which is considered to be significantly weaker and leads to finding adversarial examples at a much higher distance to the original samples. We evaluate unlearning 10% and 50% of the training samples in CIFAR-10 from a trained ResNet-18 model. As the results show, in both settings of unlearning (with access to $\mathcal{D}_R$ and no access to $\mathcal{D}_R$), using the weaker attack does not perform as well as the original method. However, it still outperforms prior SOTA unlearning methods.

# G. Comparison Using Prior Evaluation Methods

In this section we perform similar comparisons to what we presented in section 6.1, but instead of `FT AUC`, we use the same MIA used by prior SOTA methods in unlearning for evaluations. As mentioned in section 5.2, we refer to the score derived by this MIA as `MIS`.

| | RANDOM FORGET (10%) | | | | | RANDOM FORGET (50%) | | | | |
| | UNLEARN ACC | RETAIN ACC | TEST ACC | MIS | AVG. GAP | UNLEARN ACC | RETAIN ACC | TEST ACC | MIS | AVG. GAP |
|---|---|---|---|---|---|---|---|---|---|---|
| **RETRAIN** | 94.49 ±0.20 | 100.0 ±0.00 | 94.33 ±0.18 | 12.53 ±0.32 | 0.00 | 92.09 ±0.37 | 100.0 ±0.00 | 91.85 ±0.33 | 16.78 ±0.37 | 0.00 |
| **FT** | 95.16 ±0.29 | 96.64 ±0.25 | 92.21 ±0.27 | 11.33 ±0.35 | 1.84 ±0.10 | 94.24 ±0.30 | 95.22 ±0.31 | 91.21 ±0.33 | 12.10 ±0.72 | 3.06 ±0.24 |
| **RL** | 99.22 ±0.19 | 99.99 ±0.01 | 94.10 ±0.11 | 10.94 ±0.45 | 1.64 ±0.19 | 92.98 ±1.07 | 94.83 ±1.04 | 89.19 ±0.74 | 12.48 ±0.90 | 3.29 ±0.04 |
| **GA** | 98.94 ±1.39 | 99.22 ±1.31 | 93.39 ±1.18 | 4.21 ±5.25 | 3.62 ±1.04 | 99.94 ±0.09 | 99.95 ±0.08 | 94.36 ±0.31 | 0.62 ±0.30 | 6.64 ±0.15 |
| **BS** | 99.14 ±0.31 | 99.89 ±0.06 | 93.04 ±0.14 | 5.50 ±0.39 | 3.27 ±0.13 | 100.00 ±0.00 | 100.00 ±0.00 | 94.62 ±0.08 | 0.40 ±0.05 | 6.77 ±0.03 |
| $l_1$-**SPARSE** | 94.29 ±0.34 | 95.63 ±0.16 | 91.55 ±0.17 | 12.03 ±1.92 | 2.26 ±0.26 | 92.63 ±0.13 | 95.02 ±0.10 | 89.56 ±0.08 | 12.03 ±0.39 | 3.14 ±0.17 |
| **SALUN** | 99.25 ±0.12 | 99.99 ±0.01 | 94.11 ±0.13 | 11.29 ±0.56 | 1.56 ±0.20 | 95.69 ±0.80 | 97.26 ±0.79 | 91.55 ±0.59 | 11.27 ±0.94 | 3.06 ±0.12 |
| **AMUN** | 95.45 ±0.19 | 99.57 ±0.00 | 93.45 ±0.22 | 12.55 ±0.08 | **0.59** ±0.09 | 93.50 ±0.09 | 99.71 ±0.01 | 92.39 ±0.04 | 13.53 ±0.19 | **1.37** ±0.07 |
| **AMUN**$_{+SalUn}$ | 94.73 ±0.07 | 99.92 ±0.01 | 93.95 ±0.18 | 14.23 ±0.40 | 0.60 ±0.10 | 93.56 ±0.07 | 99.72 ±0.02 | 92.52 ±0.20 | 13.33 ±0.10 | 1.47 ±0.01 |

Table 10: **Unlearning with access to $\mathcal{D}_R$.** Comparing different unlearning methods in unlearning 10% and 50% of $\mathcal{D}$. Avg. Gap (see § 5.2), with MIS as the MIA score, is used for evaluation (lower is better). The lowest value is shown in bold while the second best is specified with underscore. As the results show, AMUN outperforms all other methods by achieving lowest Avg. Gap and AMUN$_{SalUn}$ achieves comparable results.

| | RANDOM FORGET (10%) | | | | | RANDOM FORGET (50%) | | | | |
| | UNLEARN ACC | RETAIN ACC | TEST ACC | MIS | AVG. GAP | UNLEARN ACC | RETAIN ACC | TEST ACC | MIS | AVG. GAP |
|---|---|---|---|---|---|---|---|---|---|---|
| **RETRAIN** | 94.49 ±0.20 | 100.0 ±0.00 | 94.33 ±0.18 | 12.53 ±0.32 | 0.00 | 92.09 ±0.37 | 100.0 ±0.00 | 91.85 ±0.33 | 16.78 ±0.37 | 0.00 |
| **RL** | 100.00 ±0.00 | 100.00 ±0.00 | 94.45 ±0.09 | 3.06 ±0.63 | 3.77 ±0.13 | 100.00 ±0.00 | 100.00 ±0.00 | 94.54 ±0.11 | 0.40 ±0.03 | 6.75 ±0.02 |
| **GA** | 4.77 ±3.20 | 5.07 ±3.54 | 5.09 ±3.38 | 32.63 ±50.85 | 76.58 ±7.73 | 100.00 ±0.00 | 100.00 ±0.00 | 94.57 ±0.06 | 0.35 ±0.10 | 6.77 ±0.04 |
| **BS** | 100.00 ±0.00 | 100.00 ±0.00 | 94.48 ±0.04 | 1.11 ±0.30 | 4.27 ±0.07 | 100.00 ±0.00 | 100.00 ±0.00 | 94.59 ±0.03 | 0.38 ±0.02 | 6.76 ±0.01 |
| **SALUN** | 100.00 ±0.00 | 100.00 ±0.10 | 94.47 ±0.10 | 2.39 ±0.64 | 3.95 ±0.14 | 100.00 ±0.00 | 100.00 ±0.00 | 94.57 ±0.12 | 0.33 ±0.04 | 6.77 ±0.03 |
| **AMUN** | 94.28 ±0.37 | 97.47 ±0.10 | 91.67 ±0.04 | 11.61 ±0.60 | 1.61 ±0.09 | 92.77 ±0.52 | 95.66 ±0.25 | 89.43 ±0.19 | 14.13 ±0.67 | 2.52 ±0.16 |
| **AMUN**$_{+Salun}$ | 94.19 ±0.38 | 97.71 ±0.06 | 91.79 ±0.12 | 11.66 ±0.16 | **1.51** ±0.02 | 91.90 ±0.63 | 96.59 ±0.31 | 89.98 ±0.44 | 13.07 ±0.66 | **2.35** ±0.15 |

Table 11: **Unlearning with access to only $\mathcal{D}_F$.** Comparing different unlearning methods in unlearning 10% and 50% of $\mathcal{D}$. Avg. Gap (see § 5.2) is used for evaluation (lower is better) when only $\mathcal{D}_F$ is available during unlearning. As the results show, AMUN$_{SalUn}$ significantly outperforms all other methods, and AMUN achieves comparable results.

# H. Continuous Unlearning (cont.)

In § 6.3, we showed AMUN, whether with adaptive computation of $\mathcal{D}_A$ or using the pre-computed ones, outperforms other unlearning methods when handling multiple unlearning requests. Another important observation on the presented results

in Figure 2 is that *the effectiveness of unlearning decreases with the number of unlearning requests*. For the setting with access to $\mathcal{D}_R$, this decrease is due to the fact that the $\mathcal{D}_F$ at each step has been a part of $\mathcal{D}_R$ at the previous steps; the model has been fine-tuned on this data in all the previous steps which has led to further improving confidence of the modes on predicting those samples. This result also matches the theoretical and experimental results in differential privacy literature as well (Dwork, 2006; Abadi et al., 2016).

This problem does not exist for the setting where there is no access to $\mathcal{D}_R$, but we still see a decrease in the unlearning effectiveness as we increase the number of unlearning requests. The reason behind this deterioration is that the model itself is becoming weaker. As Figure 8 shows, the accuracy on the model on both $\mathcal{D}_R$ and $\mathcal{D}_T$ gets worse as it proceeds with the unlearning request; this is because each unlearning step shows the model only $2\%$ (1K) of the samples and their corresponding adversarial examples for fine-tuning. So this deterioration is expected after a few unlearning requests. So when using AMUN in this setting (no access to $\mathcal{D}_R$) in practice, it would be better to decrease the number of times that the unlearning request is performed, for example by performing a lazy-unlearning (waiting for a certain number of requests to accumulate) or at least using a sub-sample of $\mathcal{D}_R$ if that is an option.

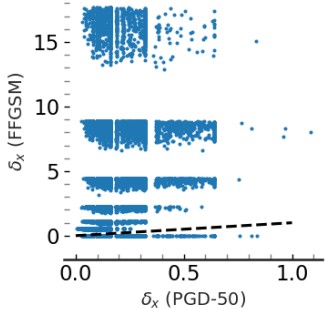

Figure 9: For each image in CIFAR-10 the $x$-axis shows the Euclidean distance of the corresponding adversarial example that is found by using PGD-50 in Algorithm 1. $y$-axis shows this distance for the adversarial examples found by the variant of FFGSM in Algorithm 1. The dashed line shows the $x = y$ line. As the figure shows, the distance is much larger for weaker attacks and this leads to worse performance of AMUN, as explained by Theorem 4.1.

