# OpenReview forum: "Not All Wrong is Bad: Using Adversarial Examples for Unlearning"
_ICML.cc/2025/Conference — ICML 2025 spotlightposter_

### Official Review · Reviewer_qDdb · 2025-03-08

**Overall Recommendation:** 3

**Summary:**

This paper proposes an algorithm for machine unlearning with an interesting finding.
The authors observe that fine-tuning models on adversarial examples closest to the corresponding forget samples can
avoid drastic changes to the global behavior of the model.

Experimental results on CIFAR-10 show promising performance compared to previous methods, like l1-Sparse, and SalUn.

**Claims And Evidence:**

The claim is empirically supported.

**Essential References Not Discussed:**

N/A

**Experimental Designs Or Analyses:**

The ablation is sound.

**Methods And Evaluation Criteria:**

(1) Fine-tuning on adversarial examples can benefit unlearning performance. This is an interesting finding. However, there is no theoretical analysis, and the experiments only focus on small data, like CIFAR-10.
It would be better to validate the generality of the finding with large-scale data, like CIFAR-100 and ImageNet1K.

**Other Comments Or Suggestions:**

N/A

**Other Strengths And Weaknesses:**

Strengths:
(1) The idea of using adversarial examples for unlearning is interesting.
(2) The proposed algorithm is simple to implement.

Other Weaknesses:
(1) The paper shows that the proposed method can also work well for the adversarially robust models which are trained with controlled Lipschitz constant.
      Currently, adversarial training is the most effective method for adversarial robustness. It uses adversarial examples as additional training data while the proposed method finetunes models on adversarial examples.
      It is interesting to know if the proposed method can work well on these models, like [ref1] and [ref2], regarding adversarial robustness with auto-attack.

[ref1] Decoupled Kullback-Leibler Divergence Loss. NeurIPS 2024.
[ref2] Better Diffusion Models Further Improve Adversarial Training. ICML 2023.

**Questions For Authors:**

See above weaknesses.

**Relation To Broader Scientific Literature:**

The paper presents an interesting finding that fine-tuning models on adversarial examples benefits the unlearning performance. It is possible to extend the idea into multi-modal models.

**Theoretical Claims:**

N/A

---

> ### Author Rebuttal · Authors · 2025-04-01
>
> We thank the reviewer for their insightful comments. We are excited about the reviewer’s acknowledgment of our interesting approach toward unlearning. Below are our responses to their questions and concerns:
>
> **Theoretical guarantees:** Although most prior SOTA methods in approximate unlearning are not accompanied by theoretical guarantees, we prove a theorem (https://shorturl.at/ChU0s) that derives an upper-bound on the 2-norm of the difference of the parameters of the unlearned model and the retrained model (which are gold-standard for unlearning). To prove this theorem, we make assumptions that are common in the certified unlearning literature. Our derived upper-bound implies enhanced effectiveness of our method when:
>
> 1. The distance between the forget sample and its corresponding adversarial example becomes smaller.
> 2. The Lipschitz constant of the model becomes smaller.
> 3. The quality of the adversarial example becomes stronger (causes a larger loss for the correct label).
> 4. The adversarial example transfers better to the retrained model.
> 5. The retrained model generalizes better to the (clean) unseen samples.
>
> Hence, the proved theorem also justifies our earlier intuitions about the need for good quality adversarial examples that are as close as possible to the original samples (which is the goal of Algorithm 1), and also justifies that by fine-tuning the model on these adversarial examples, we can derive an upper-bound on the distance between the retrained model and the unlearned one. We believe that the presented empirical results, along with the provided theorem, will motivate further theoretical studies in future work.
>
>
> **Larger model and dataset:** we have performed our experiments on VGG19 models (12 times larger than ResNet-18) trained on the Tiny ImageNet dataset (200 classes). We evaluated our prior observations similar to Figure 1 in our manuscript that shows fine-tuning the trained models on their adversarial examples does not lead to catastrophic forgetting (https://tinyurl.com/5n6f6pxr). We also compared the unlearning methods and created the tables (https://tinyurl.com/2wwwbbfp) corresponding to Tables 1 and 2 in our manuscript.
>
> In addition, as requested by another reviewer, we performed a comparison to a SOTA certified unlearning method (Zhang et al. (ICML 24)). The comparison is done only on the setting where $D_R$ is available because this method does not work when there is no access to $D_R$. Our results (https://shorturl.at/Q19RQ) show that certified unlearning methods such as this, though accompanied with theoretical guarantees, are not capable of outperforming SOTA in approximate unlearning, including AMUN. We believe that this is the case due to their assumptions not holding for deep learning models used in practice.
>
> **Adversarially-trained models:** To evaluate whether our unlearning method works with models trained using adversarial training, we performed our analysis on ResNet-18 models trained using TRADES (as it was more convenient for us to use in the rebuttal period) on CIFAR-10. We performed the experiments for unlearning 10% of the dataset in both cases where $D_R$ is accessible and not. As the results (https://tinyurl.com/43bxcafb) show, in both settings AMUN is effective in unlearning the forget samples and achieving a low gap with the retrained models. This gap is obviously smaller when there is access to $D_R$.
>
> We hope you find our responses satisfactory, and consider raising your score towards acceptance. We are happy to engage during the rebuttal period, and thank you again for your valuable comments and suggestions in improving our paper!

---

### Official Review · Reviewer_BHZY · 2025-03-13

**Overall Recommendation:** 4

**Summary:**

This paper proposes the Adversarial Machine UNlearning (AMUN) method, which reduces the prediction confidence of the model for the forget samples by fine-tuning the model on adversarial examples, while maintaining the accuracy of the model on test samples. Experimental results demonstrate that AMUN outperforms previous state-of-the-art methods in image classification tasks and performs remarkably well even in the face of membership inference attacks.

**Claims And Evidence:**

The authors clearly expound the core finding of this paper through two observations and corresponding experiments, that is, fine-tuning the trained models on adversarial examples corresponding to a subset of the training data does not lead to significant deterioration of the model's accuracy. Based on these findings, the authors propose the AMUN method, with clear logical expression and sufficient persuasiveness.

**Essential References Not Discussed:**

N.A.

**Experimental Designs Or Analyses:**

The experimental designs can demonstrate the effectiveness of the proposed method in image classification tasks. However, as the ICML is a top-tier conference in the field of machine learning, it is recommended that the authors supplement investigations on more tasks, such as text classification, to further verify the broad effectiveness of the proposed adversarial-example-based machine learning unlearning technique.

**Methods And Evaluation Criteria:**

Overall, th eproposed methods are supported by empirical data to a certain extent. The authors conducted a series of experiments on the CIFAR10 dataset for ResNet18 in the task of image classification.

**Other Comments Or Suggestions:**

Minor: In Tables 1&2, the expressions of "SALUN" and "Salun" are inconsistent.

**Other Strengths And Weaknesses:**

Strengths:

1、The proposed AMUN method is innovative. It achieves machine unlearning by fine-tuning the model with adversarial examples, opening up a new way to solve this problem. Different from previous methods, it cleverly exploits the relationship between adversarial examples and the model's decision boundary. It can not only reduce the prediction confidence of forget samples but also avoid drastic changes to the model's global behavior.

2、This paper introduces the proposed AMUN method through two observations, which is highly persuasive.

3、This paper is easy to follow.

Weaknesses:

1、The experimental designs in this paper can demonstrate the effectiveness of the proposed AMUN method in image classification tasks on the CIFAR10 dataset. It is recommended that the authors supplement the experimental results on more backbone networks (such as VGG and ViT) and more datasets (such as ImageNet). Meanwhile, as the ICML is a top-tier conference in the field of machine learning, it is advisable for the authors to conduct additional research on more tasks, such as text classification, to further verify the broad effectiveness of the proposed adversarial-example-based machine learning unlearning technique.

2、The proposed method does not seem to exhibit obvious superiority in the results presented in Tables 1&2. In particular, it is significantly weaker than the existing methods in terms of the two metrics of UNLEARN ACC and RETAIN ACC.

**Questions For Authors:**

See Weakness

**Relation To Broader Scientific Literature:**

N.A.

**Theoretical Claims:**

The method proposed in this paper is based on a heuristic methodology, and no theoretical claims are provided.

---

> ### Author Rebuttal · Authors · 2025-04-01
>
> We thank the reviewer for their insightful comments. We are excited about the reviewer’s acknowledgment of our interesting approach toward unlearning. Below are our responses to their questions and concerns:
>
> **W1. Additional experiments:** We have performed new experiments on VGG19 models (12 times larger than ResNet-18) trained on the Tiny ImageNet dataset (200 classes). We evaluated our prior observations similar to Figure 1 in our manuscript that shows fine-tuning the trained models on their adversarial examples does not lead to catastrophic forgetting (https://tinyurl.com/5n6f6pxr). We also compared the unlearning methods and created the tables (https://tinyurl.com/2wwwbbfp) corresponding to Tables 1 and 2 in our manuscript. We also performed a new experiment on the effectiveness of unlearning methods on the models trained with adversarial training, which, in summary (https://tinyurl.com/43bxcafb), shows that our method is even effective for unlearning in models trained with adversarial training. Please see the response to reviewer qDdb for details.
>
> In addition, as requested by another reviewer, we performed a comparison to a SOTA certified unlearning method (Zhang et al. (ICML 24)). The comparison is done only on the setting where $D_R$ is available because this method does not work when there is no access to $D_R$. Our results (https://shorturl.at/Q19RQ) show that certified unlearning methods such as this, though accompanied with theoretical guarantees, are not capable of outperforming SOTA in approximate unlearning, including AMUN. We believe that this is the case due to their assumptions not holding for deep learning models used in practice.
>
> **W2. Interpreting our results:** Please note that for UNLEARN ACC the goal is to minimize the difference with the corresponding value from the Retrained models. As tables 1&2 show, AMUN achieves the smallest difference in all scenarios. Similarly, for RETAIN acc in table 1 AMUN achieves the smallest difference with the corresponding value for the retrained models.
>
> For table 2, the reason that some other methods achieve smaller gap for RETAIN acc is the following: when other methods perform poorly in the absence of $D_R$, they choose the smallest available learning rate during the hyper-parameter search which basically allows them to do nothing during the fine-tuning phase. Therefore, they almost always return the same model as the original model, *without performing any unlearning*, and hence achieve an accuracy of 100% on both $D_R$ and $D_F$. But notice that in these cases, the MIA score does not decrease, reiterating that no unlearning has occurred. We will make this point clear in our future revisions.

---

> > ### Comment · Reviewer_BHZY · 2025-04-03
> >
> > Thanks for the author's response. After carefully reviewing  the author's rebuttal, most of my concerns have been addressed. I decide to maintain my score.

---

> > > ### Author Response · Authors · 2025-04-08
> > >
> > > We would like to thank the reviewer for reviewing our rebuttal and supporting our work.

---

### Official Review · Reviewer_dSmc · 2025-03-14

**Overall Recommendation:** 2

**Summary:**

The paper proposes Adversarial Machine UNlearning (AMUN), a novel method for efficient machine unlearning in classification models. The core idea is to leverage adversarial examples corresponding to the forget set to fine-tune the model, thereby reducing its confidence on $D_F$ while preserving test accuracy.  By fine-tuning on adversarial examples of the forget set (with incorrect labels), AMUN avoids global model degradation. Moreover, AMUN mimics the behavior of models retrained from scratch on the retain set to achieve comparable unlearn/retain/test accuracy and resistance to membership inference attacks (MIAs). The method also be generalized to adversarially robust models and handles continuous unlearning scenarios effectively.

**Claims And Evidence:**

"No catastrophic forgetting": The claim relies on limited datasets (CIFAR-10/ResNet-18); larger-scale experiments (e.g., Tiny-ImageNet or ImageNet) are needed for broader validation. Furthermore, it lacks theoretical guarantees.

**Essential References Not Discussed:**

Chen, M., et al. "Boundary Unlearning." (CVPR 2023): Machine unlearning is achieved by the shift decision space of the DNN model, which is quite related to adversarial example training and forgetting.

**Experimental Designs Or Analyses:**

(1) The impact of $\epsilon_{init}$ in Algorithm 1 and fine-tuning epochs is not well explored.

(2) Limited to CIFAR-10/ResNet-18; no cross-dataset/architecture validation (e.g., ImageNet, ViTs).

**Methods And Evaluation Criteria:**

(1) While PGD-50 with $l_2$ norm bounds in Algorithm 1 is logical, adaptive $\epsilon$ selection lacks theoretical justification. And the reliance on PGD-50 increases computational cost.

(2) The fine-tuning strategy is intuitive but sensitive to hyperparameters (e.g., learning rate, epochs).

**Other Comments Or Suggestions:**

N/A

**Other Strengths And Weaknesses:**

**Strengths:**

AMUN provides an alternative to machine unlearning with clear explanations of the method and its motivations.

**Weaknesses:**

The experiments are restricted to CIFAR-10 and ResNet-18 in classification tasks, raising questions about the method's applicability to larger datasets, other architectures and tasks (like Diffusion or LLM Generation). Furthermore, the paper lacks formal theoretical guarantees and thorough explorations on how variations in adversarial attack parameters affect the results.

**Questions For Authors:**

Following the previouly mentioned Strengths And Weaknesses, the questions are:

(1) Can AMUN maintain its efficacy on larger architectures (e.g., ViTs) or datasets like ImageNet?

(2) Could AMUN be combined with differential privacy or influence functions to provide certified unlearning guarantees?

(3) How does AMUN compare to certified unlearning methods (e.g., Sekhari et al., 2021) in terms of privacy-utility tradeoffs?

(4) How do choices of attack step (PGD-50 vs. PGD-10 or PGD-20), and fine-tuning epochs affect results?  Are there optimal settings for different datasets or architectures?

**Relation To Broader Scientific Literature:**

This work extends approximate unlearning methods by incorporating adversarial examples. The connections to adversarial training and Lipschitz-constrained models shows AMUN’s compatibility with robust models. Additionally, leveraging RMIA rather than MIA leads to more rigorous evaluations.

**Theoretical Claims:**

The paper does not provide formal theoretical analysis and guarantees. One of the key assumptions (e.g., adversarial examples belonging to the model’s "natural distribution") is empirically validated but lacks theoretical grounding. The relationship between adversarial example strength ($\epsilon$) and unlearning efficacy seems empirical, not theoretical.

---

> ### Author Rebuttal · Authors · 2025-04-01
>
> We thank the reviewer for their insightful comments. We are excited about the reviewer’s acknowledgment of the novel connections our work makes with adversarial training and Lipschitz-constrained models, in addition to a more rigorous analysis by leveraging RMIA rather than MIA. Below are our responses to their questions and concerns:
>
> **Q1 + W1. Additional experiments:** We have performed our experiments on VGG19 models (12 times larger than ResNet-18) trained on the Tiny ImageNet dataset (200 classes). We evaluated our prior observations similar to Figure 1 in our manuscript that shows fine-tuning the trained models on their adversarial examples does not lead to catastrophic forgetting (https://tinyurl.com/5n6f6pxr). We also compared the unlearning methods and created tables (https://tinyurl.com/2wwwbbfp) corresponding to Tables 1 and 2 in our manuscript. We also performed a new experiment on the effectiveness of unlearning methods on the models trained with adversarial training  (https://tinyurl.com/43bxcafb). Please see the response to reviewer qDdb for details.
>
> **W2. Theoretical guarantees:** Although most prior SOTA methods in approximate unlearning are not accompanied by theoretical guarantees, we proved a theorem (https://shorturl.at/ChU0s) that derives an upper-bound on the difference between the retrained models and the models unlearned using AMUN by making assumptions that are common in the certified unlearning literature. The proved theorem justifies our earlier intuitions about the need for good quality adversarial examples that are as close as possible to the original samples. For more discussions on the proved theorem, please refer to the discussion with reviewer qDdb.
>
> **Missing related work:** The work of Chen et al. is included among our baseline methods in all the experiments (BS for Boundary Shrink). We also have a thorough discussion of the differences in our approach in Appendix A.
>
> **Q2. Combination with DP:** We do not see any obvious reason why AMUN can not be combined with approaches like DP. However, AMUN exploits the robustness of the model, and this is known to have tensions with privacy (as studied in several works such as https://tinyurl.com/3573vmks, https://tinyurl.com/2n6vu3tz, https://tinyurl.com/2cdptafr). We leave a detailed investigation of this to future research.
>
> **Q3. Comparison to certified methods:** While the work of Sekhari et al. is an interesting addition to our discussion of certified unlearning methods in the related works, it is important to note that this method (similar to most other certified methods) makes many assumptions that do not hold in general deep learning models. For example, Assumption 1 in Section 4 states that “for any (input) z, the function f(w, z) is strongly convex, L-Lipschitz and M-Hessian Lipschitz with respect to w” – this is not practical. Instead, we performed a comparison to another SOTA certified unlearning method (Zhang et al ICML 24) with milder assumptions. This method does not work without access to $D_R$. Our results (https://shorturl.at/Q19RQ) show that certified unlearning methods such as this, though accompanied with theoretical guarantees, are not capable of outperforming SOTA in approximate unlearning, including AMUN, due to their assumptions not holding for deep learning models used in practice.
>
>
> **Q4. Choices of attack steps:** For the presented results on ResNet-18 and CIFAR-10, we used PGD-50. For our new results on Tiny Imagenet, we performed the experiments with both PGD-10 and PGD-20 and did not observe noticeable changes.
>
> **Q4. Instability of fine-tuning:** Please note that all the prior methods include fine-tuning steps as part of their procedure.
>
> 1. RL fine-tunes on forget samples with a random label and the remaining samples ($D_R$).
> 2. Salun does the same fine-tuning as RL, but on a subset of model parameters.
> 3. FT and l1-sparse fine-tune the model on $D_R$.
> 4. GA fine-tunes on the forget samples in the reverse direction of gradient and fine-tunes on $D_R$.
> 5. BS fine-tunes on $D_R$ and an augmented set of samples for forget samples.
>
> In our experiments, we did not observe more susceptibility to hyper-parameters compared to other methods. We also performed a fair comparison by choosing the hyper-parameters on a separate set of forget-samples and models and performed the experiments on different random seeds. To further investigate the stability of results to the number of epochs, we prepared two plots that show Avg. Gap for various number of epochs (https://shorturl.at/VSLzK). We concluded that AMUN stabilises after a few epochs and the number of epochs could even decrease, but we followed the same number of epochs that were used in prior works for fair comparisons.
>
>
> We hope you find our responses satisfactory, and consider raising your score towards acceptance. We are happy to engage during the rebuttal period, and thank you again for your valuable comments and suggestions in improving our paper!

---

> > ### Comment · Reviewer_dSmc · 2025-04-05
> >
> > I appreciate the authors' efforts in providing comprehensive analyses and additional experimental results to address the raised questions. From an empirical standpoint, the effectiveness and generalization of AMUN have been demonstrated. However, I still hold some concerns about theoretical claims and proofs. The theorem derives an upper bound on the parameter difference between the unlearned model and and the retrained model, where the bound explicitly incorporates adversarial example strength and model properties. But two issues may exist in the current theoretical part:
> >
> > (1) The derivation relies on the convexity assumption of the loss landscape, whereas neural networks inherently exhibit non-convex optimization surfaces. Such non-convexity might induce parameter convergence to local minima rather than global optima to potentially violate the theoretical guarantees.
> >
> > (2) The theorem does not theoretically justify whether adversarial examples belong to the model’s **"natural distribution"**. If adversarial examples lay outside the training data distribution, its effectiveness for unlearning may degrade due to distribution shift.
> >
> > If authors could solve my concerns, I would be very pleased to increase my score.

---

> > > ### Author Response · Authors · 2025-04-06
> > >
> > > We thank the reviewer for reviewing our rebuttal in detail and raising points that could lead to further clarification of our method. Below are our responses to their questions and concerns:
> > >
> > > **Q1:** The assumption of convexity is common in prior works related to certified unlearning:
> > >
> > > 1. The work mentioned by the reviewer (Sekhari, A. et al. (NeurIPS 2021)), makes an even stronger assumption that the model is strongly convex (assumption 1 in Section 4).
> > >
> > > 2. Later works (Chien, E., (ICLR 2023)) make a similar assumption to bound the inverse Hessian of the loss with respect to the parameters (proof of Theorem 4.3 in Appendix A.7).
> > >
> > > 3. The SOTA method in certified unlearning (Zhang, B., et al. (ICML 24)), which we used in our comparisons (https://shorturl.at/Q19RQ), uses similar assumptions on bounding the inverse Hessian matrix, but they exploit the local convex approximation to derive the bound (Lemma 3.3 in Section 3). Their theoretical guarantees are based on other assumptions as well, such as the Lipschitz continuity of Hessian of the loss (Assumption 3.2 in Section 3) , which does not hold for neural networks, and also the size of the model parameters (Theorem 3.4 in Section 3).
> > >
> > > That being said, our work is focusing on proposing an approximate unlearning method. Most prior methods in approximate unlearning, including the work mentioned by the reviewer (Chen, M., et al. (CVPR 2023)) and the current SOTA (Fan, C., et al. (ICLR 2024)) *do not provide any form of theoretical guarantees*. They only rely on empirical evaluation using (weaker, non-SOTA) membership inference attacks to verify the effectiveness of their method. It is noteworthy to say that although these methods lack theoretical guarantees they lead to better results than the certified methods in practice.
> > >
> > > Not only are our results superior to those reported here, we also provide some analysis theoretically. This in itself exceeds the contributions made in several published papers, some of which have received spotlights.
> > >
> > > While there is a mismatch between the theoretical frameworks used for analyses and practical deployments, these guarantees still provide useful information and hypotheses about the relevant factors influencing the quality of the unlearning methods in simpler settings. We believe that they will also motivate future research on extending the theoretical guarantees to the more general settings.
> > >
> > > **Q2:** First, we would like to clarify that by mentioning that *"the adversarial examples belong to the natural distribution learned by the trained model"*, we do not mean that they belong to the distribution of the training data (which itself is an ongoing research thread on characterizing adversarial examples that are on-manifold or off-manifold for the underlying data distribution, e.g, https://tinyurl.com/2sw3vbcn). The adversarial examples that we compute are specific to the model. Once a model $M$ is trained on the training data, it imposes a distribution on the set of all possible samples. For a sample $(x,y)$ that belongs to the forgetset we find a perturbed version of $x$ i.e., $x’$, for which the model makes the prediction $(y’ \neq y)$. Therefore, from the model’s perspective (learned distribution by the model), the correct label for $x’$ is $y’$, which means that $(x’, y’)$ belongs to the *distribution that the model has imposed on the set of all possible samples*. Hence, although $y’$ is the wrong prediction for the sample $x’$, it matches the distribution learned by the model. However, this does not necessarily hold for a separately trained model because the sample $(x’, y’)$ is specifically crafted for model $M$.
> > >
> > > To empirically evaluate that fine-tuning the model on adversarial examples does not lead to a distribution shift in the distribution that the original model imposes on the input space, we have plotted the confidence values on the test set and the remaining set before and after unlearning with AMUN. The results can be found here: https://tinyurl.com/4fp87wnj . As the resulting violin plot shows, after using AMUN the distribution of the confidence values for these subsets of the input space barely changes.
> > >
> > > Thanks again, for your questions and engagement. We are happy to answer any more questions you have.

---

### Official Review · Reviewer_fXrz · 2025-03-15

**Overall Recommendation:** 3

**Summary:**

This article introduces AMUN, an unlearning method that uses adversarial examples to remove the influence of specific training samples from a trained model while preserving overall model accuracy. The key insight behind AMUN is that fine-tuning a model on adversarially modified versions of the forget set (DF) enables effective unlearning without requiring full retraining.

**Claims And Evidence:**

The authors claim that AMUN Effectively Unlearns Data Without Significant Accuracy Drop
This is supported by: Tables 1 & 2 (AMUN achieves lower Avg. Gap than baselines). Still no comparison of computational efficiency.

Claim: AMUN Works Even Without Access to Retained Data (DR)

This is supported by: Table 2 (AMUN+SalUn achieves lowest Avg. Gap). But no experiments on larger datasets or different data types is performed.

Claim: AMUN Works for Adversarially Robust Models

This statement is supported by: Table 3 (AMUN achieves similar performance to retraining). But no analysis is shown on how robustness constraints are affecting the unlearning process.

Claim: AMUN Supports Continuous Unlearning

This is supported by: Figure 2 (AMUN-A outperforms other methods across multiple steps). But no discussion on computational cost of sequential unlearning is given by the authors.

Problematic or Unclear Claims

1. Computational efficiency: Claimed but not directly compared to baselines.
2. Scalability: No experiments on large models/datasets.
3. Generalization to other architectures: Only tested on ResNet-18 (CIFAR-10).

**Essential References Not Discussed:**

NA

**Experimental Designs Or Analyses:**

1. The article benchmarks AMUN against various unlearning baselines (e.g., Retrain, FT, RL, GA, BS, SALUN).
The results show AMUN’s advantage, statistical significance tests (e.g., hypothesis testing) are not reported, making it unclear if improvements are meaningful beyond noise.


2. The article evaluates performance in different settings (random forget 10% & 50%, adversarially robust models, continuous unlearning).
The continuous unlearning scenario (AMUN-A) is promising but lacks a long-term stability analysis—effects of multiple iterations on overall model performance are not well examined.

**Methods And Evaluation Criteria:**

1. Proposed Method (AMUN) uses Adversarial Examples for unlearning few samples.

2. Effectiveness Measured by Key Metrics

Test Accuracy (Test Acc) – Ensures model retains performance.
Forget AUC (FT AUC) – Measures how well forgotten data is removed.
Avg. Gap – Lower values indicate better unlearning.

3. Comprehensive Baseline Comparisons

Evaluated against Retrain, FT, RL, GA, BS, SALUN, AMUN+SalUn.

4. Tested on CIFAR-10 Dataset but authors should try experimenting with CIFAR-100 and other segmentation datasets as well.

5. Authors should try evaluation on deep architectures like ViTs or large-scale models.

**Other Comments Or Suggestions:**

My Suggestions:

1. The paper lacks rigorous theoretical guarantees for why adversarial fine-tuning effectively unlearns data.
2. Performance relies on the quality of adversarial examples, which can be computationally expensive and can go over several iterations. How does this guarantee that an adversarial example of a given sample will really unlearn? Have the authors conducted an analysis on gradient based maps or loss landscapes?
3. Results are primarily shown for CIFAR-10; effectiveness on larger, more complex datasets is unclear.
4. While various unlearning methods are tested, more recent and advanced SOTA techniques could have been included.
5. Authors should discuss about the limitations of this method.
6. A good block diagram is missing.
7. Some tables are too small to be seen properly. I can understand the space issues but a good adjustment of important tables should be done (eg. Table 3)
8. 5.2. Evaluation Metrics para can be shortened to save space.

**Other Strengths And Weaknesses:**

Strengths:

1. Authors introduce adversarial fine-tuning as an effective unlearning method. This is unique perspective in this field.
2. AMUN has an edge over existing unlearning methods in Avg. Gap and MIA resistance across various settings.
3. Avoids costly full retraining while maintaining competitive accuracy.
4. Demonstrates effectiveness in iterative unlearning.

Weaknesses:
1. The paper lacks rigorous theoretical guarantees for why adversarial fine-tuning effectively unlearns data.
2. Performance relies on the quality of adversarial examples, which can be computationally expensive and can go over several iterations. How does this guarantee that an adversarial example of a given sample will really unlearn? Have the authors conducted an analysis on gradient based maps or loss landscapes?
3. Results are primarily shown for CIFAR-10; effectiveness on larger, more complex datasets is unclear.
4. While various unlearning methods are tested, more recent and advanced SOTA techniques could have been included.
5. Authors should discuss about the limitations of this method.
6. A good block diagram is missing.
7. Some tables are too small to be seen properly. I can understand the space issues but a good adjustment of important tables should be done (eg. Table 3)
8. 5.2. Evaluation Metrics para can be shortened to save space.

**Questions For Authors:**

Concerns:

1. Your method relies on adversarial fine-tuning to approximate unlearning. Do you have any theoretical guarantees that AMUN effectively removes information from the model rather than just masking it? If not, how do you ensure true unlearning?

2. Given the computational cost of generating adversarial examples, how does AMUN scale to large-scale datasets like ImageNet? Have you tested its efficiency and effectiveness in such settings?

3. The results suggest AMUN reduces MIAs, have you tested it against stronger, adaptive attacks designed specifically to bypass adversarial fine-tuning? If so, how does it perform compared to retraining?

4. Your comparisons include various unlearning techniques, but how does AMUN perform against more recent methods such as certified removal techniques or data augmentation-based approaches? Would AMUN still hold its advantage in Avg. Gap and AUC metrics?

5. How will the users determine the forget set?

6. The results in the table 1 show incremental pattern and the proposed method does not show any significant accuracy change.

7. Performance relies on the quality of adversarial examples, which can be computationally expensive and can go over several iterations. How does this guarantee that an adversarial example of a given sample will really unlearn? Have the authors conducted an analysis on gradient based maps or loss landscapes?

Authors can discuss about some of the above points rather than experimenting further.

**Relation To Broader Scientific Literature:**

This article builds on prior works in machine unlearning, particularly efficient unlearning methods that avoid full retraining.

**Theoretical Claims:**

1. This article lacks a formal proof of complete data removal.

2. No proof linking AMUN directly to better generalization. I suspect this is of utmost importance and the authors should consider showing this proof. This is an intuitive claim, but no theoretical analysis of effectiveness.

3. Observation 1 is very trivial.

---

> ### Author Rebuttal · Authors · 2025-04-01
>
> We thank the reviewer for their detailed and insightful comments. We are excited about the reviewer’s acknowledgment of the uniqueness of our approach for unlearning. Below are our responses to their questions:
>
> **Q1+S1+W1. Theoretical guarantees:** Although most prior SOTA methods in approximate unlearning are not accompanied by theoretical guarantees, we proved a theorem (https://shorturl.at/ChU0s) that guarantees moving toward the retrained models by making assumptions that are common in the certified unlearning literature. Please refer to the discussion with reviewer qDdb for more details.
>
> **Q2+S3+W3. Larger model/dataset:** we have performed our experiments on VGG19 models (12 times larger than ResNet-18) trained on the Tiny ImageNet dataset (200 classes). We evaluated our prior observations similar to Figure 1 in our manuscript that shows fine-tuning the trained models on their adversarial examples does not lead to catastrophic forgetting (https://tinyurl.com/5n6f6pxr). We also compared the unlearning methods and created the tables (https://tinyurl.com/2wwwbbfp) corresponding to Tables 1 and 2 in our manuscript. Please see the response to reviewer qDdb for more details.
>
> **Q4. Certified baseline:** We performed a comparison to a SOTA certified unlearning method (Zhang et al ICML 24). This method does not work when there is no access to DR. Our results (https://shorturl.at/Q19RQ) show that certified unlearning methods such as this, though accompanied with theoretical guarantees, are not capable of outperforming SOTA in approximate unlearning, including AMUN. We believe that this is the case due to their assumptions not holding for deep learning models used in practice.
>
> **Q3. Stronger attacks:** We wish to clarify that the objective of our algorithm is to use adversarial attacks to find "neighbors" for those samples in the unlearning set, and use this neighbor set for fine-tuning to ensure that the resulting (unlearned) model lacks confidence on these samples. We are unsure of what the reviewer means to consider attacks to bypass adversarial fine-tuning, as the objective of the attack we consider is to "augment" the sample (to be unlearned) in a manner so as to ensure that it's confidence is low post fine-tuning.
>
> **Q5. Forget set:** Upon receiving an unlearning request, the set of samples to be forgotten are specified. This is the common setting used in prior works in unlearning for classification models, such as SulUn and l1-sparse.
>
> **Q6. Table 1:** The setting of Table 1 (access to DR) is much easier than Table 2. Still, we want to point out the fact that even in that scenario, our method achieves the smallest Avg Gap (based on the average over 27 runs). Moreover, once we make things more difficult, by either moving to a larger model and dataset, or revoking access to DR, the advantage of our method over prior works becomes apparent. The capability of performing unlearning without access to DR is much more desirable as it will be more practical in many real-world use-cases.
>
> **Q7+S2+W2. Computational cost:** The time comparison: https://shorturl.at/AXVB7 . Note that, we only need to run Algorithm 1 on the samples that are requested to be forgotten. For our experiments, we choose a small sub-sample of the corresponding dataset and evaluate their final values of $\epsilon$; based on these values, we set the initial value of $\epsilon$ and run PGD attack on the samples. Then we only keep the samples for which an adversarial example is not found, and run another round of PGD with the updated $\epsilon$ value. We proceed until all the samples find one corresponding adversarial example. These histograms (https://shorturl.at/fMQDC) show the number of samples for each $\epsilon$ value. Based on our analysis, for both CIFAR10 and tiny imagenet, finding the adversarial example for all the samples is equivalent to less than 3 runs of PGD50 and PGD20, respectively.
>
> **W4+S4. More recent SOTA:** SalUn is a SOTA method in approximate unlearning published in ICLR 24. In our experiments we added all the baseline methods from that work. In addition, we also performed experiments using the SOTA in certified unlearning (Zhang et al ICML 24), which has made our baselines comprehensive and up-to-date.
>
> **W5+S5. Limitations:** We have tried various settings, such as Lipschitz-bounded models and adversarially-trained models to ensure our approach is not limited to only regular training paradigms, but there is no guarantee on compatibility with all training paradigms. Also, as with every new method, the introduction of this new methodology for unlearning might invite a certain line of attacks specifically targeted for this approach. Also future approaches are required to avoid the slight degradation in adaptive setting.
>
> We hope you find our responses satisfactory, and consider raising your score towards acceptance. We are happy to engage further, and thank you again for your valuable suggestions in improving our paper!

---

> > ### Comment · Reviewer_fXrz · 2025-04-02
> >
> > I am happy with the clarifications given by the authors. I raise the score to 3.

---

> > > ### Author Response · Authors · 2025-04-08
> > >
> > > We would like to thank the reviewer for reviewing our rebuttal and supporting our work by updating their score.

---

### Decision · Program_Chairs · 2025-05-01

**Decision:**

Accept (spotlight poster)

**Comment:**

The papers proposes a novel unlearning method, AMUN, that achieves unlearning by fine-tuning on adversarial examples (based on the forget set). The method is simple yet effective, and works without access to the retain set, and can handle "iterative" unlearning. The paper is very well written, with a clear motivation behind the method, and theoretical analysis. During the rebuttal, the authors addressed reviewer concerns, and added additional experiments that included evaluation on larger models/datasets, new SOTA methods, and tests on adversarially trained models. The proposed method AMUN seems to perform well when evaluated using standard unlearning metrics, and strong MIAs. In general, as recognized by the reviewers, it seems like a promising approach that others can build on.